# A Generalist Intracortical Motor Decoder

## Abstract

Mapping the relationship between neural activity and motor behavior is a central aim of sensorimotor neuroscience and neurotechnology. Most progress to this end has relied on restricting complexity: studying specific simple behaviors, in limited subjects, with interpretable computational models. However, current trends in deep learning suggest that modeling a breadth of neural and behavioral data all at once is not only possible, but that such a model would also benefit downstream analysis of related data. We accordingly developed Neural Data Transformer 3 (NDT3) as a foundation model for motor decoding of neural data from intracortical microelectrodes. We pretrained NDT3 with 2000 hours of neural population spiking activity paired with diverse motor covariates from over 30 monkeys and humans from 10 labs. Pretrained NDT3 is broadly useful, benefiting decoding on 8 downstream decoding tasks and generalizing to a variety of neural distribution shifts. However, we find signs that scaling over diverse neural datasets may be challenging, as scaling from 200 to 2000 hours already requires increasing model size to 350M parameters to avoid model saturation, and several downstream datasets scarcely benefit from scale. We provide two demonstrations that this scaling is at least partially limited by variability in input and output spaces across neural datasets, which pretraining alone may not resolve.

## 1 Introduction

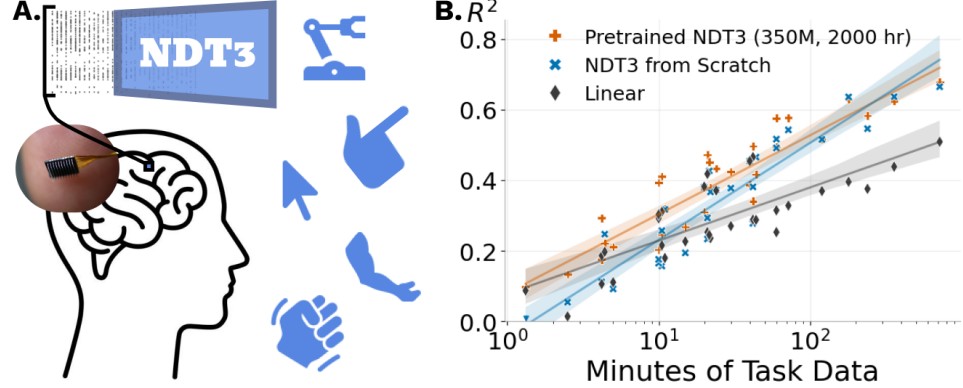

**Figure 1. A.** NDT3 is a deep network for decoding intracortical spiking activity into low-dimensional time series for various motor effectors[1]. **B.** We aggregate decoding performance on downstream tasks with variable amounts of data (from Fig. 11). While from-scratch models only reliably outperforms a linear baseline after 15 minutes of data, tuning a 350M param. NDT3 pretrained with 2000 hours of data is consistently better at all scales.

Intracortical neural data collection is growing rapidly. This growth comprises not only larger individual datasets with more neurons and higher behavioral complexity (Urai et al., 2022; Stevenson, 2023), but also an increase in the collective number of datasets. This wealth of data presents an opportunity to develop insights and applications that span multiple datasets at once, provided we can reconcile their inherent diversity. Large deep networks appear very suitable for this task, so much

---

[1]Photo courtesy of REDACT and The Chicago Tribune.

so that the creation of deep networks operating on broad domain data has been termed foundation modeling (Bommasani et al., 2022). Efforts to create foundation models are now proliferating beyond their origins in natural language processing (NLP) and computer vision (CV) into many domains of engineering and science (Wang et al., 2023c). Here, we propose a foundation model for motor decoding from intracortical spiking activity, which we call Neural Data Transformer 3 (NDT3).

Motor decoding is a valuable initial domain for characterizing neural data foundation models. Academic, clinical, and industrial efforts to create iBCIs for neuroprosthetics provide a path for scaling data collection from hundreds to thousands of subject-hours, and also fuel a need for pretrained models that generalize quickly and perform robustly for new users and settings. Behavior prediction metrics for BCI performance are also more intuitive for benchmarking progress than neural data prediction or the abstract goal of providing scientific insight (e.g. with latent variable models or in silico models) (Pei et al., 2021; Wang et al., 2023b). Finally, recent work has shown that deep networks are able to transfer learn across motor cortical datasets collected at different timepoints, subjects, or tasks (Azabou et al., 2024; Ye et al., 2023; Schneider et al., 2023). These ingredients provide the motivation and means for scaling neural data modeling.

However, scaling may be constrained by the design and heterogeneity of modern neural datasets. By design, we refer to the fact that many neural datasets restrict behavioral complexity to probe specific hypotheses. These restrictions impoverish not only the behavioral signals but also the observed neural data (Gao and Ganguli, 2015), providing us a narrow window through which to understand the general relationship between neural activity and behavior. Beyond the limitations of individual datasets, each neural dataset inherently contains unique variability distinguishing them from others. This is most salient when comparing across the datasets aggregated in pretraining, where different neurons are recorded in each subject and distinct output dimensions are required for each effector. To illustrate why these factors together challenge scaling, consider a 2-neuron toy setting, where both neurons fire noisily. One neuron fires on leftward motion and the other fires on rightward motion. No amount of scaling on other datasets could reduce the data needed from this setting to determine which each neuron's preferred direction, but neither is the problem trivial due to stochastic firing.

To assess the value of scaled pretraining on heterogeneous spiking activity, we developed Neural Data Transformer 3 (NDT3). NDT3 uses simple tokenization strategies to enable pretraining over diverse datasets and fine-tuning to new tasks without introducing any new parameters (Fig. 1A). We pretrained NDT3 using up to 2000 hours of neural and behavioral data from motor neuroscience experiments with monkeys and clinical iBCI trials with humans. We then evaluated NDT3's decoding performance on eight diverse motor tasks (Section 3.1) and find that tuning NDT3 yields models that either improve or match task-specific models trained from scratch, with prominent gains when task data is under 1.5 hours (Fig. 1B). Further, these gains persist under a number of distribution shifts (Section 3.3). These benefits may enable both more complex experimental design and potentially decrease the burden of decoder training for people using iBCIs. However, our results also suggest that neural data heterogeneity may be limiting scaling. Scaling pretraining data to 2K hours required raising model capacity to 350M parameters to mitigate performance saturation, and some tasks accrue no benefits from scale at all. We identify NDT3's sensitivity to the specific inputs and outputs seen during fine-tuning as two limits to be overcome for more productive neural data foundation models.

## 2 APPROACH

### 2.1 DATA

NDT3 models datasets of paired neural spiking activity and behavior (Fig. 2). Given our focus on motor decoding, most of the data comes from devices implanted in motor cortex of various monkeys and humans (Fig. 2A). These devices are intracortical multielectrode arrays or probes that record 30 kHz extracellular potentials. Spikes are extracted from these potentials, typically by bandpass-filtering the data between 300 and 3000 Hz, and marking a spike when the voltage signal crosses a preset threshold value. The neural data in our pretraining are diverse (Fig. 2B top). Data can have markedly different profiles across electrodes due to being from different electrode arrays in the same subject (left), have many silent channels (middle), or be densely active due to noise (right).

The typical behaviors in the pretraining data are different types of upper-limb reaching and grasping, nearly all from experimental paradigms that consist of short, repeated trials. While neural data were

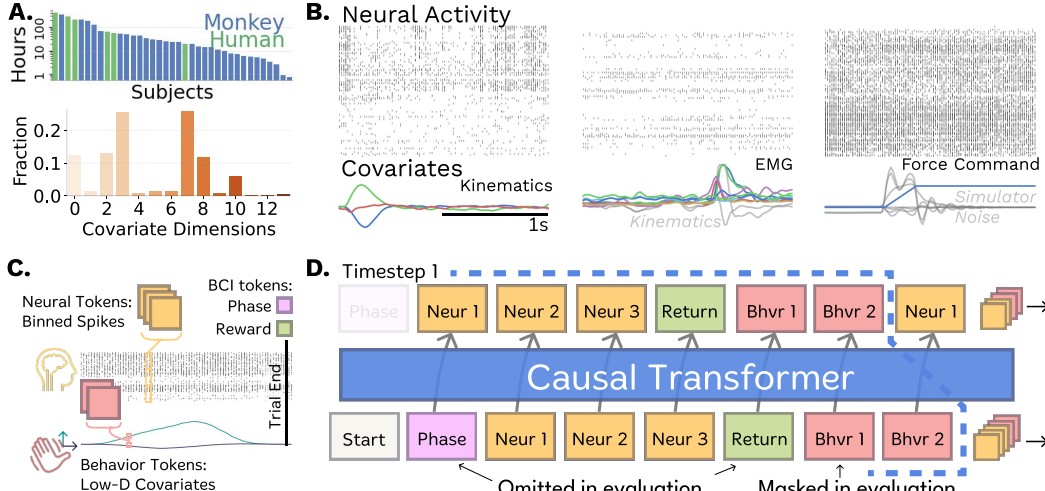

**Figure 2. NDT3 Data and Model Design: A.** NDT3 models paired neural spiking activity and behavioral covariate timeseries. We plot the distribution of 2000 hours of pretraining data volume by subjects (top) and covariate dimensionality (bottom). **B.** Examples of the neural and behavioral data for each of the three types of behavioral covariates in pretraining: kinematics, EMG (electromyography), or forces. Not all modeled dimensions in data are meaningfully task-related (right, grey behavior). **C.** Neural spiking activity is tokenized in time by binning the number of spikes every 20 ms, and in "space" using patches of channels (usually 32), as in NDT2 (Ye et al., 2023). Behavior is low-dimensional in our data, so we use 1 token per behavior dimension, also per 20 ms timestep. NDT3 also pretrains on data from BCI control, which we annotate with two additional tokens. The phase token indicates whether the user is controlling or observing the behavior and the reward token indicates if the BCI task was completed. **D.** NDT3 models tokens in a single flat stream with linear readins and readouts. Every real-world timestep (shown by the blue cutout) yields several tokens, which we order to allow causal decoding in evaluation. In evaluation, we omit return and phase tokens and zero-mask behavior tokens.

always recorded from microelectrodes, motor covariate signals came from various sensors. In monkey datasets, these sensors measure actual limb activity (e.g., Fig. 2B, left: limb kinematics from optical tracking, middle: electromyography (EMG)). In human datasets, physical movements are typically not possible, so the data's behavior signals are programmatically generated. These signals are "paired" with the neural data in that they are cued or otherwise instructed to the person, who will attempt or imagine the corresponding behavior, such as grasping at a specified force level (Fig. 2B right). This force panel also shows that in pretraining, we cannot always automatically discern the primary task covariates (e.g., blue line, force, in the panel) from other recorded behavioral variables (grey). Thus, some behavior variables may unpredictable. Finally, we include closed loop iBCI data, where some behavior is generated by an iBCI decoder (not NDT3, see modeling strategy in Section 2.2).

The pretraining datasets are composed of archives from several experimental labs and some public datasets, and contain data from non-human primate neuroscience experiments and human clinical trials for neuroprosthetics. The grassroots nature of this aggregate dataset presents a heterogeneity in neural data processing, motor effectors, and experimental setup, most comparable to aggregate robotics datasets (OpenX et al., 2024). We detail the pretraining data composition in Section C.4.

We minimize preprocessing of these data to maximize the applicability of our generalist model. Kinematic signals are typically converted to velocities, and all behavior (kinematics, EMG, force) is normalized per dataset such that the maximum absolute value of each variable is one. Data are cut or concatenated into fixed length sequences, without additional annotation of data discontinuity. This strategy, common in language modeling (Geiping and Goldstein, 2022), homogenizes the data for improved GPU utilization while maintaining throughput of real data. We used a length of two seconds as it is roughly the timescale of the behavior in our data (Fig. 2A). Sequences with no spikes or covariate variability are discarded. In total, this yielded about 3 million sequences, 1 billion neural tokens, or 1750 hours, which we sample uniformly for pretraining. We round this to 2 khrs in subsequent text for simplicity.

## 2.2 MODEL

NDT3 is a causal Transformer with linear readin and readout layers for its various modalities, similar to GATO or TDMs (Reed et al., 2022; Schubert et al., 2023; Chameleon, 2024). For use with a Transformer, the data must be tokenized (Fig. 2C). We tokenize neural data by patching spike counts (Ye et al., 2023); each token is a flattened vector of the binned spikes in a chosen temporal resolution (20 ms) and spatial dimension (32 channels). For example, neural activity sampled from an electrode array with 100 channels would patch into $4 = \lceil 100/32 \rceil$ 32D neural tokens per 20 ms timestep. As the behavioral variables are already low-dimensional, we simply assign 1 token per dimension at the same temporal resolution as neural data. Finally, we add tokens marking whether the behavior are generated by a BCI system or by physical limb movement. While measured kinematics, EMG, or force will reflect a natural relationship with neural activity, behavioral data from BCI tasks are controlled by a program or learned decoder. We frame BCI-driven behavior as a suboptimal demonstration (Merel et al., 2016), and adopt a scheme inspired by Decision Transformers (Chen et al., 2021; Lee et al., 2022). In this scheme, we use a Phase token to track the timesteps where behavior is at least driven by neural activity and under decoder control, or only under programmatic, open loop control. We also use a Return token reflecting controller quality based on task completion. Note that these signals are only considered for pretraining, and are ablated entirely from the model at evaluation. Similarly, input behavior tokens are masked out in inference, so that the model input only indicates how many tokens must be predicted. NDT3 is trained with mean-squared error for prediction of behavioral variables, and categorical cross-entropy losses for prediction of neural spike count and reward.

All modalities are flattened into a single token stream, with the order of tokens in each real-world timestep respecting a canonical order required for control (Fig. 2D). As in GATO, individual tokens are annotated with learned position embeddings identifying token modality and sub-modality "position." We additionally use rotary embeddings (Su et al., 2023) to track real-world timesteps.

**Pretraining and Fine-Tuning** We pretrain NDT3 models over variable pretraining data and in sizes of 45M and 350M parameters to assess the impact of data and model scaling. Pretraining is early stopped according to validation loss or terminated at a maximum of 400 epochs. The 200 hour, 45M model trains for 480 A100-hours while the 2000 hour (2kh) 350M model takes 20K A100-hours. Fine-tuning maintains the pretraining objectives and updates all parameters.

## 2.3 EVALUATION STRATEGY

**Evaluation datasets and tuning** Our main evaluation (Section 3.1) uses four human and four monkey datasets sampling varied upper limb movements, which we detail in Section C.4. Each dataset contains multiple sessions of data, typically from a single monkey or human. We will refer to each such setting as a "task," distinguished from the behavioral procedure performed in each dataset. Each session has unique variability, so fine-tuning procedure may greatly impact decoding results. Prior work (Azabou et al., 2024; Ye et al., 2023; Zhang et al., 2024) ran focused evaluations by tuning and evaluating separately for each evaluation session. To manage compute and storage demands and to reflect that real world datasets are rarely collected or analyzed in isolation, we fine-tune NDT3 jointly over data combined from multiple evaluation sessions. Fig. 12 shows this joint tuning outperforms focused tuning for multi-session data.

**Baselines** We compare against Wiener filters (WF) and NDT2. WFs are a conventional linear method for both motor decoding and control in iBCI devices (Pandarinath and Bensmaia, 2022), and we implement them as ridge regression with multi-timestep history. NDT2 is a Transformer that uses MAE-style (He et al., 2021) self-supervision to learn across multiple neural datasets. We detail the differences between NDT2 and NDT3 in Section C.3. We compare to NDT2 prepared both from scratch and tuned from the public checkpoint pretrained on 100 hours of human data. Note for tractability we – Other Transformers have been proposed for scaling over spiking data (Azabou et al., 2024; Zhang et al., 2024), but the field yet lacks consensus benchmarks to evaluate these proposals.

**Downstream Hyperparameters** We tune all deep networks (NDT2 and NDT3) over 3 learning rates. This sweep is limited to make computational demands tractable, but also demonstrates the versatility of the base model. Importantly, the same search space is used for all tasks; we list the space and show its sufficiency relative to wider sweeps in Section C.2. The best learning rate is chosen based on average validation score over three random seeds, and we report their mean on the evaluation data.

## 3 RESULTS

NDT3's pretraining effort advances prior intracortical models an order of magnitude in data and model scale, from 200 to 2000 hours and 10M+ to 100M+ parameters. In Section 3.1, we show the increased data scale saturates aggregate downstream performance unless simultaneously increasing model scale. We propose that the performance drop from scaling data alone is due to high variability across intracortical motor decoding datasets. In Section 3.2, we show how this variability reflects in NDT3's sensitivity to shifts in data input or output. Section 3.3 concludes by showing that despite this challenge for further data scaling, NDT3's pretraining already provides gains that generalize to various novel settings, establishing NDT3 as a useful foundation for motor decoding.

### 3.1 MULTI-SCALE EVALUATION ACROSS MOTOR DECODING TASKS

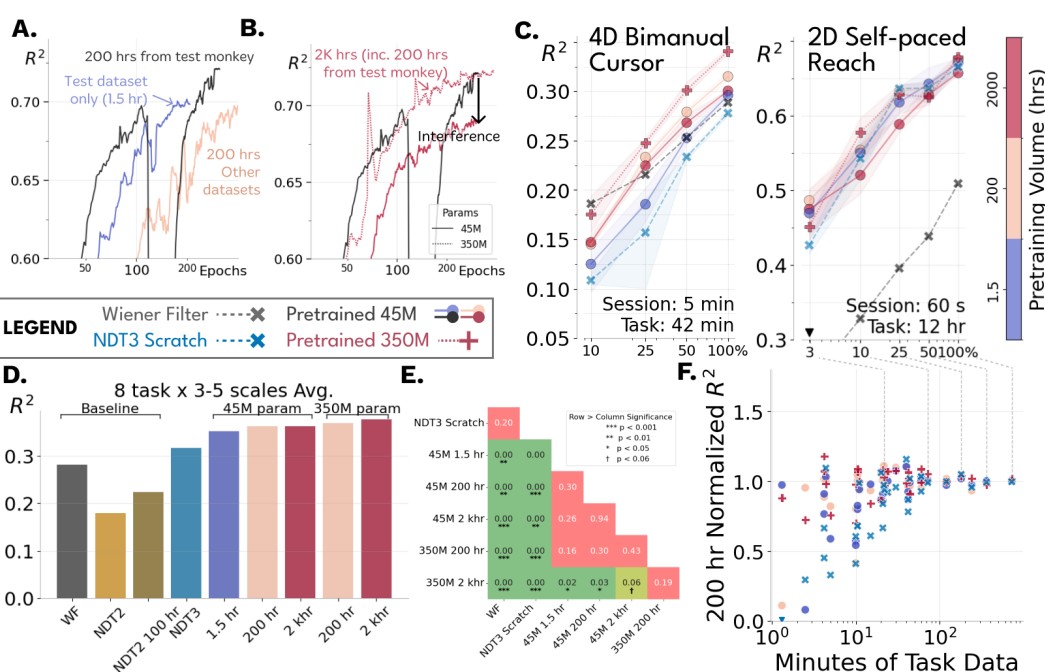

**Figure 3. Evaluation on diverse motor tasks:** A single legend and color scheme is used throughout. **A.** Test-split pretraining $R^2$ compared for 3 models. All model pretraining data includes 1.5 hours of calibration data for the test dataset. We compare a model with just this data (Test dataset only) vs using 200 hours of additional data either from the test monkey or from over 10 other monkeys. Only the additional test monkey data improves over the calibration model. Models terminate at different points due to early stopping. **B.** Pretraining $R^2$ for models with up to 2000 hours (2 khrs) of pretraining data. The 2 khr model degrades in performance vs the 200 hr model at 45M parameters and merely maintains performance at 350M parameters. **C.** Examples of good and bad data-scaling in downstream multiscale evaluation on two datasets. The bottom right text shows time in each evaluation session and total time in each dataset. The x-axis scales this full dataset down by random subsampling. Shading shows standard deviation on 3 tuning seeds. Increasing pretraining data yields performance gains at all downstream scales in the 4D task, but effects are unclear in the self-paced reach task. ▼ indicate outliers clipped for clarity. **D.** Downstream performance averaged for 31 settings comprised of different downstream datasets and scales, for different NDT3s and baselines. 45M NDT3s improve with data from 1.5 hrs to 200 hrs but saturate at 2 khrs. Increasing model size to 350M parameters enables further gain. **E.** p-values computed from FDR-corrected pairwise t-tests for each pair of models. The 350M 2 khr NDT3 significantly outperforms other pretrained NDT3s, except the 350M 200 hr NDT3, and is the only model to do so. NDT2s omitted for brevity, see Fig. 13. **F.** Per-task performance, normalized by the 350M 200 hr NDT3 performance, is shown against task time for different NDT3 models. Each vertical band shows models trained on the same evaluation setting, e.g. dashed lines show the evaluations from the self-paced reaching dataset. Model variability vanishes by 1.5 hours.

To set expectations for how data scale and model size will impact model performance, we first examine pretraining curves computed on a test split. This test split contains data from multiple sessions of 2D reaching behavior mainly from one monkey. Models are able to learn this test task in pretraining as they are given 1.5 hours of data separately sampled from these sessions. Fig. 3A

compares the test performance of a model using just this 1.5 hours of calibration data with those of two 200 hour models. The "Other datasets" 200 hour model used data from 10 other monkeys performing a variety of reaching tasks, but did not improve over the minimal 1.5 hour model on the test data. In contrast, using 200 hours from the test monkey (from a separate set of experiments with similar behavior) achieved a small improvement in performance. Thus, only closely related data appears to benefit a model that already uses sufficient task-specific data, in this case 1.5 hours. Fig. 3B further shows that scaling to 2 khr can actually degrade test performance, suggesting dissimilar pretraining data can interfere with learning of the test task. This interference is mitigated by increasing model size to 350M parameters, consistent with prior recommendations to scale model size and dataset size in tandem (Dosovitskiy et al., 2021; Kolesnikov et al., 2020; Aghajanyan et al., 2023). However, the test task does not improve beyond what is achieved by providing 200 hours from the test monkey.

This upstream saturation motivated a downstream evaluation conducted at multiple data scales. We illustrate this evaluation for two tasks in Fig. 3C. These two datasets are from a human performing open loop iBCI calibration for bimanual cursor use (Deo et al., 2024), and a monkey performing self-paced reach to random targets (O'Doherty et al., 2017). In both cases, the individuals are held-out from pretraining entirely, so the task-specific data is only seen in tuning. In the bimanual task, NDT3 performance improves with increased pretraining data at all downstream data scales. Encouragingly, and distinct from the saturation seen in pretraining, joint scaling of data and model size to 2 khr and 350M parameters improves performance in the bimanual task. The self-paced reach dataset shows a much less clear result. For example, the from-scratch NDT3 is competitive at all data scales.

These tasks show just two of several different trends on the eight evaluation datasets we study. We defer discussion of individual tasks and their variability to Section B.5, and next consider summary performance. Fig. 3D was produced by tuning over 2000 models in 31 evaluation settings, and identifies an overall benefit of pretraining scale. To begin, NDT3 from scratch outperforms the WF and NDT2, whether pretrained or not. We discuss NDT2's poor performance in Section B.7. This from-scratch NDT3 performance can be improved with minimal pretraining (1.5 hrs), consistent with findings in computer vision (Entezari et al., 2023), and continues improving up to 200 hours of pretraining data. Further scaling to 2000 hours is also helpful, but only when paired with increased model size to 350M parameters, as in upstream evaluation. The gain of the 350M 2 khr model over other models is statistically significant for all except the 350M 200 hr model (Fig. 3E). Other pairs of pretrained NDT3s are not significantly different at the current scale of evaluation.

While scaling shows an overall positive trend, benefits vary greatly across downstream settings. While we have already seen that comparing benefits by experimental task is challenging, it is known that pretraining is most beneficial at low downstream data scale. We confirm this in our setting by plotting normalized task performance against downstream scale in Fig. 3F. As in unnormalized performance (Fig. 1B) and upstream results (Fig. 3B), the distinction between pretrained models and from-scratch models vanishes by 1.5 hours, or 0.5-5K data points. Vanishing benefits by 5K data points, while far from trivial, implies that NDT3's pretraining is relatively impotent by CV or NLP standards (Kolesnikov et al., 2020; Wang et al., 2019), and in practical terms can be exceeded after a few sessions of data collection. Importantly, since scaling *downstream* data past 1.5 hours still reliably improves performance, we cannot attribute the saturation of pretraining to insufficient signal in each dataset's neural activity.

## 3.2 PRETRAINING IS LIMITED BY INPUT AND OUTPUT VARIABILITY IN TUNING

In Section 3.1 we observe that the benefit of scaling pretraining appears to saturate at relatively low downstream data scales. It is possible that this limit reflects intrinsic variability across neural datasets, with a long tail of specialized features that are needed for the best performance in each dataset. Alternatively, our modeling decisions around architecture, hyperparameters, and post-training, may all have significantly limited NDT3's scaling. As a first step towards dissociating these factors, we next analyze NDT3's sensitivity to the specific neural inputs and covariate outputs seen in tuning.

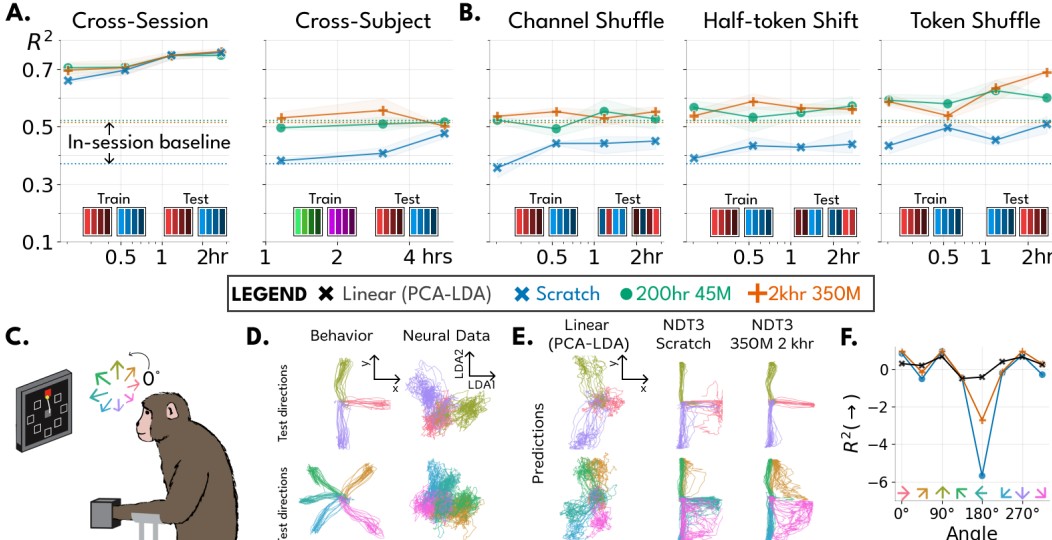

**Figure 4. NDT3 fails in certain novel input and output configurations. A: Cross-session transfer persists after pretraining, but cross-subject does not.** We test NDT3 in a downstream task with one evaluation session from a monkey self-paced reaching dataset. Training uses 1 minute from the evaluation session and additional data from other sessions (Cross-Session) with the same monkey or from sessions from a different monkey for the same behavior (Cross-Subject). **B: Shuffling inputs ablates cross-session data to resemble cross-subject transfer.** uses the same cross-session neural data but permutes input dimensions (recording channels). Shuffle channel randomly permutes inputs, half-token shift rolls channels so that each channel $i$ uses data from $i + 16$, and shuffle token permutes data patchwise, keeping channels from the same patch together. Channel shuffling and half-token shifts both are sufficient to reduce cross-session transfer to the same level as cross-subject transfer. All panels show the baseline performance achieved by the model with just 1 minute of test-session data, x-axis shows additional cross-context data provided. **C-F.** Pretraining does not improve angular extrapolation. **C.** We study a new dataset where a monkey performs an isometric center out task, with exerted forces mapped to cursor positions in 8 different angles. **D.** We split data into three held-in and five held-out angles. Behavior is cleanly separated across conditions. The neural data for each condition can also be visualized separably by projecting them to 2D plane computed by combining PCA and LDA. **E.** Predictions derived either by fitting a Wiener Filter to the projected neural data (Linear, PCA-lDA), or from NDT3 (Scratch, 350M 2khr). While the linear model generalizes to held-out angles, NDT3 predictions are restricted to held-in ranges. **F.** Pretraining quantifiably improves over from-scratch in all conditions, but far underperforms the generalization of PCA-LDA.

**Input order sensitivity may limit cross-subject transfer.** In neural data, the effectiveness of transfer learning is greatly reduced when using cross-subject data compared to cross-session data (Ye et al., 2023). This suggests limited scaling may be caused in part by the vanishing utility of cross-subject data, even when the data is collected in identical experimental setups and thus controlled for other variables. We can illustrate this by comparing cross-session and cross-subject transfer after large-scale pretraining. On a monkey 2D reaching dataset (O'Doherty et al., 2017) in Fig. 4A, we tune NDT3 with calibration data from one test session and additional cross-session or cross-subject data. As in Section 3.1, cross-session data is still highly beneficial even after pretraining, but cross-subject data is only helpful for from-scratch models. Pretrained models in the cross-subject setting instead begin and plateau at a performance that is just slightly better than the best cross-subject performance in from-scratch models. These results suggest that NDT3's pretraining has already learned the features that cross-subject transfer provides in this task, supporting the idea that scaling is limited by a low (task-dependent) ceiling on cross-subject transfer.

This low transfer stems from cross-dataset variability, but may be exacerbated by NDT3's design and inductive bias. To dissect NDT3's sensitivities, we observe that since cross-subject data contain different neurons, cross-subject transfer must at least accommodate changes in the specific semantics of each data dimension, a problem which we term "sensor variability." We can isolate how effectively NDT3 resolves sensor variability by transferring with cross-session data while permuting the test-session's neural dimensions. Fig. 4B Channel shuffle shows that input permutation cripples the ability of NDT3, whether pretrained or not, to learn from cross-session data. We next apply structured ablations of input order, as in Neyshabur et al. (2020). We find that even the small alteration of a half-token shift in channel order is sufficient to reproduce the effect of full shuffling (Fig. 4B center).

This shows NDT3's cross-session transfer depends greatly on the specific token dimension semantics, and may drive the observed limits in scaling. Finally, we consider a shuffle that only alters the test session's neural token order with respect to cross-session data, hypothesizing Transformers can more easily correct token ordering alterations. Indeed, pretrained and from-scratch models recover some gains when cross-session data is only token-shuffled (Fig. 4B right). This manipulation shows the influence of model design on transfer.

For reference, POYO (Azabou et al., 2024) adopts a graph-based Perceiver design motivated by this sensor variability challenge, which makes it a promising candidate to scale. However, their experiments only show nontrivial cross-subject transfer rather than robustness to input perturbations as we conduct here. Since NDT3 also achieves cross-subject transfer, stronger evidence is needed to support that Perceiver-style models may scale better. Finally, we note that to control for potential confounds from tuning over heterogeneous data, we evaluated both sequential and joint tuning strategies in these experiments and reported the better approach in each panel. A full comparison is given in Section B.6.

**Pretraining does not enable angular extrapolation**. The increased data efficiency of pretrained models suggests that NDT3 could decode a new subject's behavior without sampling the full range of neural and behavioral data. For simple behaviors, this expectation provides a validation of whether pretraining has succeeded at all. Center-out reaching provides this simple litmus test as its underlying neural activity can be visualized in a planar subspace (Churchland et al., 2012). This visualization doubles as an explicit prior for decoders aiming to generalize to held-out angles. We expect that NDT3's pretraining could learn this prior, enabling generalization to reach directions yet unseen in a *new* subject. Note that non-pretrained deep networks fail at such held-out angle generalization (Rizzoglio et al., 2022). To evaluate this hypothesis, we analyze a single session of an isometric monkey center-out dataset, where the monkey exerts forces in one of eight angles and its force level is mapped to cursor position (Fig. 4C). We separate this data into 3 held-in and 5 held-out angles, as shown in Fig. 4D. We can visualize the corresponding separability of the neural data by first applying principal components analysis (PCA) and then linear discriminant analysis (LDA, see Section B.1 for methods). These held-out neural data are still organized radially, by reach angle (Fig. 4D bottom right).

This reduced view of the neural data implies we can build a linear decoder that generalizes to held-out angles. Indeed, a Wiener Filter on this 2D neural data predicts behaviors that, while generally low in quality, are coarsely aligned with the correct reach angle (Fig. 4E). In contrast, NDT3 from scratch predictions for held-out angles are constrained to their nearest held-in angles. Pretraining provides mild improvements for the interpolated angles at $\pm 45°$, but largely replicates the failure to predict extrapolated angles. We quantify prediction accuracy in Fig. 4F, showing that while pretrained NDT3 improves over from-scratch NDT3 overall, pretraining still far underperforms a simple prior in generalizing to held-out angles. We repeat this evaluation in two more settings in Section B.1. Importantly, this demonstration leaves open the question of whether pretraining has failed to learn the utility of dimensionality reduction, or whether aligning NDT3 to yield behavioral generalization in tuning will require further model post-training.

These two studies on model input and output highlight the difficulty of pinpointing whether NDT3 scaling is limited by data or methodology. However, they do provide basic tests of model capability, namely robustness to channel shifts and generalization to unseen behaviors, that we expect future approaches will need to overcome to achieve better neural data foundation models.

### 3.3 WHERE DOES PRETRAINING HELP?

Despite challenges for scaling further, NDT3's pretraining has learned a prior from hundreds of hours of neural data. We next provide examples where this pretraining does usefully generalize.

**Neural distribution shifts** Neural data is nonstationary, with shifts rising from a mix of controlled experimental variables to more speculative factors. For example, the firing rate of different channels will evolve over the course of an hour, implying a distribution shift associated with change in time (Fig. 5A top left). Shifts also occur between activity in two arm postures or whether finger motion occurs under spring load or not (Fig. 5A top middle and right). Since these shifts are common in neural data, pretraining gains should ideally be robust to their effect. We thus tune models on data from one setting (in-distribution, ID) measure the performance of models in that same setting and the

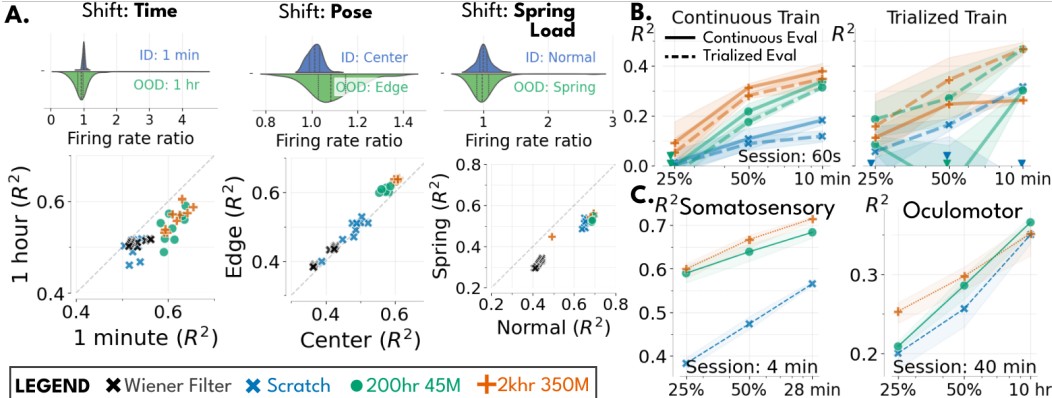

**Figure 5. Generalizability of pretraining gains. A.** Models fine-tuned in one distribution of data are evaluated in-distribution (ID) and out-of-distribution (OOD). Top plots show the distribution across channels of neural firing rates from OOD and ID trials, normalized by average ID firing rates. Lower plots scatter OOD vs ID performance, with each point being a single model with different hyperparameters. . The **time** shift uses two human cursor datasets collected one hour apart. Models were tuned in each block and were evaluated in the second block. **Pose** shift uses a monkey center-out reach task which was performed with the hand starting in different locations in the workspace. **Spring Load** uses a dataset of monkey 1D finger motion with or without spring force feedback. **B.** Models are evaluated on a human open-loop cursor dataset prepared in two ways. Trialized training receives inputs according to trial boundaries, varying from 2-4 seconds in length. Continuous training receives random 1 second snippets (that can cross trial boundaries). Trialized evaluation matches trialized training, and continuous evaluation is done by streaming up to 1 second of history. ▼ indicates points below 0.0. Continuously trained models perform well in both evaluation settings, while models trained on trialized data fail in continuous evaluation. **C.** Multiscale fine-tuning performance of NDT3 on datasets recorded outside motor cortex, namely S1 (Somatosensory) and FEF/MT (Oculomotor).

shifted setting (out-of-distribution, OOD). Positive correlation of performance in all cases imply the ID gains conferred by pretraining persist OOD. This ID-OOD correlation is consistent with Miller et al. (2021), implying a potentially fruitful relationship between the distribution shifts characterized in neural data and those studied in computer vision. More practically, these examples suggest that pretraining benefits are not dependent on narrow features specific to the choice of tuning dataset.

**Trial structure** DNNs have been observed to overfit the temporal structure of experimental data, challenging their use in iBCI control (Deo et al., 2024; Costello et al., 2023). For example, a DNN might learn there is always no motion before the start of a behavioral trial, independent of the neural activity. To date, these claims have been studied exclusively on un-pretrained deep networks. We next assess how pretraining affects this overfitting in open loop human cursor control data by comparing a continuous and trialized setting. In the continuous setting, we cut random one second intervals of data in training and continuously stream up to one second of data in evaluation. The trialized setting formats data to respect trial boundaries, so the model always sees data aligned to the start of behavior.

In Fig. 5B, we show that models using both trialized training and evaluation outperform models with both continuous training and evaluation. This implies NDT3 will learn to exploit clear trial structure in data. However, while trialized from-scratch models become subtrivial under continuous evaluation (solid blue line is off-panel), pretrained models degrade more gracefully. For example, the 350M 2 khr model evaluated continuously only performs slightly worse with trialized tuning than with continuous tuning. Pretraining NDT3 thus reduces its dependence on trial structure, which should benefit both data analysis and iBCI control. Note however the contrast in these results with Fig. 4C, which show that DNNs clearly do overfit to tuning data in some cases. These nuances underscore the importance of rigorously evaluating model generalization in future work.

**New brain areas** In Fig. 5C, we return to multiscale fine-tuning to test how NDT3, pretrained on motor cortex, performs in somatosensory cortex (S1) and oculomotor areas (FEF and MT). Pretraining provides a large boost over from-scratch models is high in S1, but also nontrivial in the Oculomotor dataset. While the former can be attributed to the close interaction of sensorimotor areas, the latter implies NDT3 has learned a broader prior. While encouraging, our results thus far suggest this prior could reflect neurophysiology (e.g. declining subject focus over time (Steinmetz et al., 2019)), but

might also reflect common experimental artifacts like trial structure. For example, this Oculomotor dataset contains 4 behavioral conditions, which may benefit from the tendency to learn classifiers shown in Fig. 4C rather than a prior on neural dynamics.

## 4 DISCUSSION

Many fields are now pursuing large scale deep learning as "a tide that lifts all boats" (Abnar et al., 2021), with the hope that improvements in effective pretraining will yield field-wide, downstream improvements. Such a unifying abstraction may be timely for neuroscience, given the increasing volume, diversity, and complexity of modern neural data. Joining other pretraining efforts on varied modalities of neural data (Section A), we trained NDT3 on 2000 hours of paired neural population activity from motor cortex and behavior, and then conducted a broad downstream decoding evaluation. Consistent with the broad foundation modeling narrative, we found the best aggregate performance from increasing data scale and model size jointly (Section 3.1). However, these benefits from pretraining vary with the downstream dataset, with several datasets having minimal improvements from scale (Section B.5). This result may stem in various ways from our approach, for example in insufficient breadth of hyperparameter sweeps, or too narrow of a focus on decoding metrics. Alternatively, we have highlighted how improving downstream gains may require new architectural innovations robust to input ordering shifts, and possible new training strategies to promote generalization (Section 3.2). Overall, NDT3 establishes a strong baseline foundation model for intracortical decoding from spiking activity, but highlights important directions for future scaling.

More broadly, we advocate for further consideration of how neural data can contribute to and gain from the ongoing cross-disciplinary conversation on foundation modeling. For example, our input and output sensitivity analyses were inspired by ML (Neyshabur et al., 2020; Pham et al., 2021) and neuroscientific literature (Gallego et al., 2020; Sadtler et al., 2014), respectively. Improving scaling in neural data may benefit from insights developed from characterizing multimodal interference more broadly (Aghajanyan et al., 2023; Liu et al., 2024). Inversely, neural distribution shifts have the advantage of being carefully studied, and so the appearance of correlated ID-OOD performance in neural data, as also appears in CV, NLP, and other AI domains (Taori et al., 2020), may refine our understanding of when such correlation will occur, and thus when foundation models will be effective. Our hypothesized challenge of sensor variability should be particularly interesting to compare across the biosignals community, which must overcome analogous variability to achieve our shared goal of achieving user-general models.

### 4.1 ETHICS STATEMENT

The animal datasets used in this work were collected for other studies that were approved by Institutional Animal Care and Use Committees. Human datasets were also collected for other studies, with Institutional Review Board approval and as part of clinical trials conducted under FDA Investigational Device Exemptions. Informed consent was obtained prior to any experimental procedures. We discuss the potential for NDT3 to reduce user burden for iBCI-based neuroprosthetics, though the dissemination of pretrained models on these data raise the risk that the original human data may be recoverable from model weights. Since this seems technically challenging at this point, and since the source data are restricted to binned spiking activity to begin with, we deem the risk low enough to justify the potential scientific benefit of sharing our pretrained models.

### 4.2 REPRODUCIBILITY STATEMENT

Advancing neural data foundation modeling will require a flourishing open-source ecosystem, including data, models, and evaluations. While we will release our models and codebase, our work currently has limited reproducibility given our inability to release pretraining data. Similarly, we have tried to use open evaluations where possible, but several evaluation datasets remain private. We expect that field-wide trends toward open data releases, and larger scale academic (Koch et al., 2022) or academic-industrial collaborations, can alleviate this limitation in the near future.

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

## A    RELATED WORKS AND A PROPOSED TAXONOMY

Neural data is sufficiently diverse so as to support many distinct efforts to train large neural data models. The scale of pretraining is somewhat larger in the non-implanted modalities, where data is more abundant. The largest EEG models have reached a scale of 2.5K (Jiang et al., 2024) to 40K hours of data (Yuan et al., 2024), or higher volumes if also considering non-brain biosignals (EKG) (Yang et al., 2023; Thapa et al., 2024). Current fMRI models operate in the 1K (Thomas et al., 2023) to 7K (Caro et al., 2024) hour range. The largest models in these studies are in the 0.1B-1B parameter range. Intracranial modalities, including sEEG (Wang et al., 2023a; Chau et al., 2024), ECoG (Talukder et al., 2022; Peterson et al., 2022), and spiking activity (Wang et al., 2023b), have thus far been studied at an order of magnitude smaller scales of data and model size (20-1000 hours, <0.1B parameters).

Direct scaling on neural data modeling should be distinguished from NeuroAI efforts (Doerig et al., 2022) to measure how models of the human sensorimotor experience (e.g. language, vision, audio models) predict neural data (Antonello et al., 2024). However, as multimodal efforts begin to blur this distinction (Benster et al., 2024; Xia et al., 2024), care will be required to distinguish advances in modeling neural data, embodied data, or their interaction.

**Comparing neural data models** Current efforts to understand scaling in neural data Simeon et al. (2024); Sato et al. (2024) will have their reach limited by the specificity of every neural dataset. A meta-challenge for the field is understanding how different parameters (species, brain area, modality, task) impact scaling properties. This would be greatly aided by development of reporting practices for different neural data models. To facilitate comparison, we create a model card (Mitchell et al., 2019) for NDT3 in Section D. In addition to the standard model card, we propose reporting an additional taxonomy to aid comparisons across neural data models, using two concepts.

First: neural data models can be conceptualized as modeling slices of the *plenneural function*, inspired by the plenoptic function in vision (Adelson et al., 1991). The plenoptic function is a model of an idealized eye which parameterizes all possible images with 7 dimensions: 4D to describe the global spacetime of the view, 2D to describe viewing angle (spherical) or coordinate (Cartesian) of the image, and 1D for wavelength. Since neural data models are primarily interested in circumscribed systems rather than the physical world, a similar global coordinate system (e.g. 4D for all possible electric potentials) would be uninformative. We thus propose reporting more qualitative coordinates:

1. Identity: The network or individual being recorded.
2. Task: The behavior, stimuli, or other activity the network is reflecting.
3. Spacetime: Coordinates specified in a network-local coordinate frame (e.g. brain area).

Second: The modeled extent of this plenneural function is conveniently discretized in three resolutions in a Transformer-like sequence modeling framework: the token, the sequence, and the full training data. The token is the most granular unit of data being modeled; NDT3 models neural populations 32 neurons at a time, in 20ms bins. At the sequence input level, NDT3 models inputs from single humans or monkeys, across 128-256 neurons in 2 second snippets, while performing effectively one "movement." Finally, NDT3's pretraining spans dozens of individuals, records motor and premotor areas over 2.5K hours, over a variety of arm and hand movements.

## B    SUPPLEMENTARY RESULTS

### B.1    FURTHER TESTS OF HELD-OUT ANGLE GENERALIZATION.

To further support our single-dataset illustration of attractor structure in pretrained NDT3, we evaluate reach angle generalization across sessions in the isometric, force-based dataset (Monkey J) setting and a second, manipulandum-based (Monkey J) setting with monkey movement. To begin, we visualize the separability of these neural data with LDA (Fig. 6A Neural Data). We next train decoders on every pair of angles separated by 90 degrees (one shown) and plot predictions on held-out trials from all angles. NDT3 and WFs, here directly fit to high-D data instead of after PCA-LDA, both fail to extrapolate to held-out angles, consistent with Rizzoglio et al. (2022). We quantify prediction performance in Fig. 4D. These plots again show that held-out generalization is subtrivial, while being entirely consistent with pretraining's overall narrative of improved performance in all conditions.

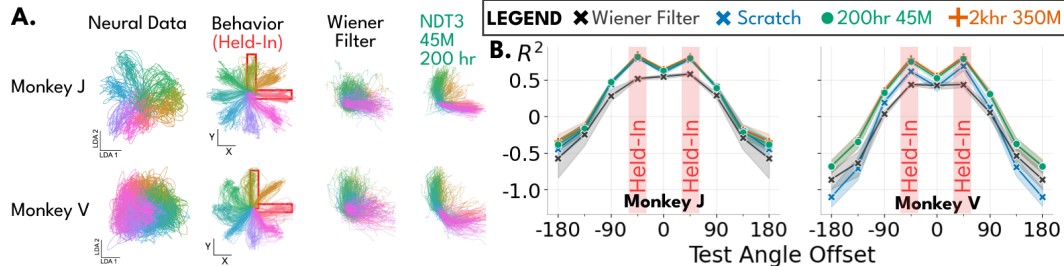

**Figure 6. A.** Two monkey 2D center-out datasets with 8 angular conditions. LDA projections show monkey J's data is distinctly more separable than monkey V's. We then test generalization of behavioral decoding to held-out angles after training on 2 of 8 angles (boxed in red). All NDT3s and the WF produce predictions constrained between the held-out angles. **B.** We quantify performance for decoding on each angle with respect to distance from the central angle between the held-in angles. We average performance over all 8 central angles.

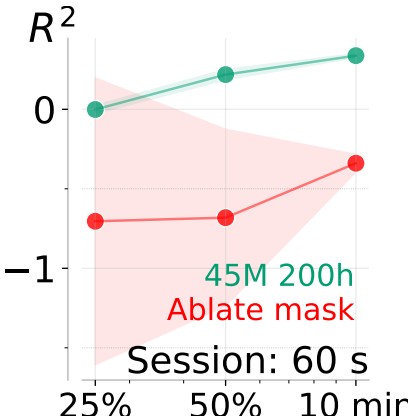

**Figure 7.** Ablation of covariate masking on an open 2D Cursor + Click dataset. Covariate inputs are completely masked in inference for the default NDT3, and autoregressively generated in the ablation.

**Methods for PCA-LDA.** Our PCA-LDA preparation used in Fig. 4E-F adheres closely to our standard data preparation and is directly comparable to the inputs received by the deep networks. To begin, we smooth all spike counts with an exponential kernel, as in our Wiener Filter baseline (Section C.3). We then fit PCA to the high-dimensional (96D) neural activity at each timestep and extract the top 10 PCs. Finally, we fit linear discriminant analysis (LDA) to reduce this 10D PC space to neural activity to 2D. The training data for PCA and LDA are both restricted to only the held-in angles. Note that LDA uses categorical labels to separate the 3 held-in reach directions but is applied without class labels after fitting. While this train-time labeling is technically not given to NDT3, this does not influence our argument that NDT3 pretraining should be capable of angular generalization, as NDT3 does not confuse the reach angle of held-in predictions. The final Wiener Filter is fit to the 2D data and uses 1 second of history.

## B.2 ABLATIONS

We ablate the major design decisions made to enable NDT3's large scale pretraining. These ablations give us confidence that NDT3 overcomes the basic challenges we encountered in development, but compute restrictions prevent more exhaustive comparisons or exploration of model design space. We encourage further work exploring the influence of different hyperparameters. In these plots, we distinguish validation split performance and evaluation split performance, which is computed by batch-mode prediction (not the costly streaming evaluation used throughout main experiments).

**Covariate dropout** We find the default next-step prediction objective fails for learning decoding of highly autocorrelated covariate timeseries, perhaps because simply relying on teacher-forced behavioral inputs provides a severe shortcut that prevents learning of a proper neural to behavior decoding map (Bachmann and Nagarajan, 2024). Different time-series models have addressed

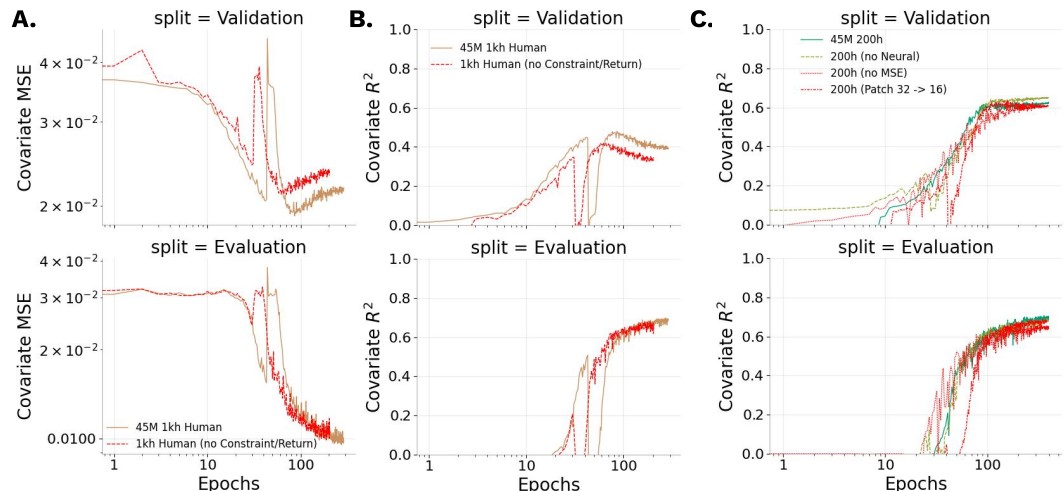

**Figure 8.** Ablations evaluated based on upstream evaluation split. **A., B.** Ablation of BCI control tokens **C.** Ablation of neural objective and covariate MSE objective in favor of classification over quantized covariates.

this by adopting convolutional input-output layers (Lea et al., 2016), tokenizing along temporal dimensions (Das et al., 2024), or learning with contrastive objectives (Chau et al., 2024; Kostas et al., 2021). We avoid introducing architectural modifications and instead adopt a simple dropout procedure that masks a portion of covariate inputs some fraction of the time. Specifically, on every training batch, two random numbers are drawn. The first, $M \sim U[0, 1]$, determines what fraction of covariate inputs should be masked. On 90% of batches, we also sample $T \sim U[0, 2]$ seconds, such that the mask is only applied after timestep $T$. That is, on 90% of batches, the model is provided a prefix-prompt. We do not block losses on this prefix as in prefix-LMs (Wang et al., 2022). Pretraining metrics for validation and evaluation are always computed with a prefix and full masking of non-prefix timesteps. In Fig. 7, we ablate covariate masking (which also removes the prefix logic), and tune on a 2D Cursor + Click task. The ablated model performs subtrivially with student-forced predictions provided as input at test time. Note that the ablated model performs trivially with masked inputs (not shown).

**BCI-phase and return conditioning** NDT3's pretraining includes several hundred hours of BCI control data, where the covariates were set by another decoder. We introduced phase and return conditioning tokens to differentiate the several types of BCI control data from recorded behavior. Specifically, in BCI data, NDT3 receives input tokens specifying what fraction of the behavior reflects neural input (BCI control is on) vs programmatic input (BCI control is off, as in open loop BCI calibration). Further, we provide inputs encoding reward (trial success) when trials change, and return (future reward over a 10 second horizon, which crosses data boundaries). This design is intended to evaluate the potential for a Decision-Transformer like offline learning strategy for improved online control, but we do not discuss this in this work. In Fig. 8A, B, we focus on whether these inputs improves pretraining loss and $R^2$ in validation splits, which has contains BCI data, and the held-out evaluation split containing only monkey behavior. The figures show that the ablation significantly decreases validation split performance, and causes a slightly earlier stopping point leading to worse evaluation performance. Note both models early before the full training budget of 400 epochs.

**Neural reconstruction objective** All main NDT3 models used a neural reconstruction objective inherited from the self-supervised learning pretraining from NDT2. We ablate this choice post-hoc and see it may actually minorly harm pretraining (validation split), though the neural objective doesn't harm evaluation split decoding (Fig. 8C). Note the scalar weighting of neural vs covariate objectives were set to be roughly balanced in pretraining. Section B.3 provides a downstream analysis on the standalone value of the neural reconstruction objective.

**MSE over classification** In robotics and certain generalist models (Schubert et al., 2023), continuous action spaces are sometimes better decoded and controlled when quantized (Shafiullah et al., 2022). This is because MSE is an insufficient objective when the output distribution is multimodal (e.g. one of two possible paths in robotics). While it seems unlikely that the close relationship between

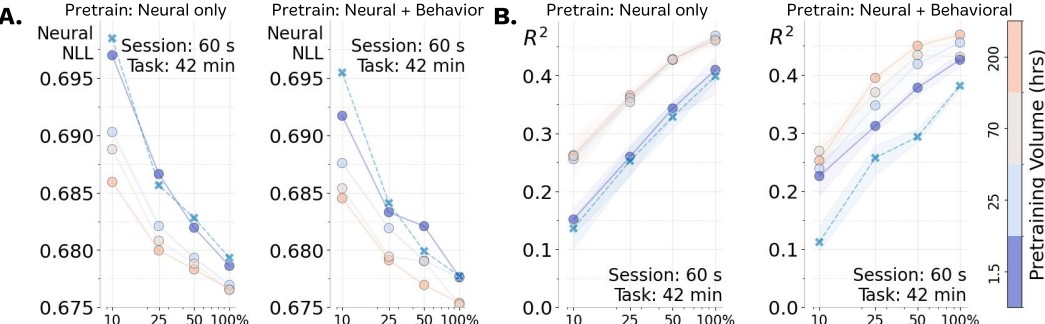

**Figure 9. Scaling of unsupervised pretraining** After pretraining models up to 200 hours with only the neural reconstruction objective, we fine-tune models in a similar multiscale evaluation as in Section 3.1, with a newly initialized behavioral readout. We use the CST task here. **A.** left shows neural loss scales with neural-only pretraining. The right panel plots the neural loss, also present in the standard evaluation, also scales with joint pretraining. **B.** left shows that with neural pretraining, downstream decoding performance saturates at a flat performance after 25 hours. This is compared against the nonsaturated scaling from joint pretraining. Colorbar is common for all plots, and Xs are from-scratch NDT3 models.

movement behavior and motor cortex is multimodal, multimodal behavior may be appropriate when pretrained on heterogeneous data, i.e. when similar neural activity corresponds to different behavior in two datasets. We attempted such a quantization, including HL-Gauss smoothing (Farebrother et al., 2024) which we found to help; but this does not recover the performance of the default MSE objective (Fig. 8C) on the evaluation split. We found this performance gap persisted under fine-tuning (not shown). This suggests that NDT3 is differentiates neural data inputs from different datasets.

**Patch size** NDT2 and NDT3 both tokenize neural data by patching them into fixed size clusters. It is unclear whether transfer learning might occur for sub-token features, which motivates the use of smaller tokens in larger datasets that might afford it (Caron et al., 2021). We change patch size to 16 and show this performs slightly worse in the 45M 200h model (Fig. 8)C. Smaller patches (and subsequent increased neural tokens) may be more beneficial in the larger scale models, but their benefit must be weighed against their increased compute burden.

### B.3 ISOLATED SCALING IN NEURAL DATA

Due to NDT3's joint modeling of behavior and neural data, it is difficult to dissociate whether scaling gains in behavior come from improved behavioral or neural priors. To assess whether NDT3 can scale solely from neural data modeling, we pretrain a new set of 45M parameter models up to 200 hours with only causal neural data modeling objective. As before, we then tune to a downstream decoding task, in this case, a Critical Stability Task dataset (CST). From a representation probing perspective, improved downstream performance implies higher quality neural representations. We use the standard single-stream autoregressive modeling objective as in the rest of this work in the downstream setting, we find direct linear probing of neural representations perform worse. Fig. 9 compares the scaling on downstream neural and behavioral metrics after the standard fine-tuning procedure.

Fig. 9A shows that downstream neural reconstruction improves with increased pretraining data either when using only the neural objective or both neural and behavioral objectives (as in the standard setting). The joint pretraining achieves advances neural metrics in all settings, illustrating that decoding behavior is a complementary objective to neural data reconstruction even for representation learning.

Fig. 9B contrasts decoding curves in the two pretraining settings, in that neural pretraining has saturated decoding after just 25 hours of pretraining. This is consistent with the interpretation that the behavioral readout reflects only one aspect of the neural data. Together with the neural metric plots, this analysis shows scaling over solely neural data is possible, but also that decoding behavior is a complementary pretraining objective for improving neural representation learning and decoding.

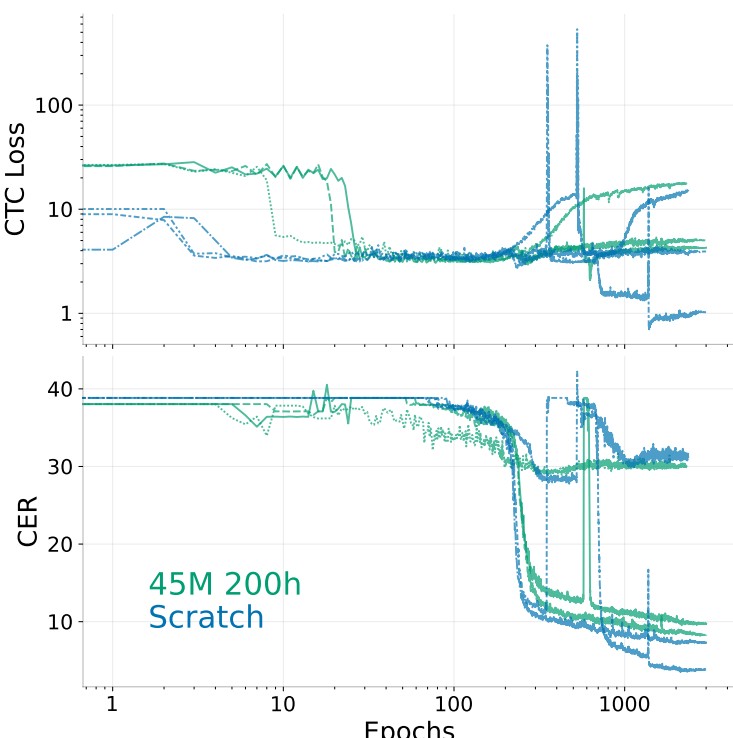

**Figure 10.** Three regimes of NDT3 training for handwriting decoding. We show validation loss and character error rates for example runs of from-scratch and fine-tuned NDT3s.

### B.4  PRETRAINING DOES NOT BENEFIT FALCON H2 (HANDWRITING)

We also evaluated NDT3 for decoding of letters in a human-open loop handwriting task (FALCON H2). Although this is also a motor cortical decoding task, we excluded H2 from NDT3's aggregate evaluation since it is a sequence-to-sequence as opposed to continuous task. To apply NDT3 to this task, we pool neural tokens at each timestep and add a linear projection and optimize with a CTC loss (Graves et al., 2006). We maintain the default neural reconstruction loss and causal attention mask, and do not apply data augmentation.

Note that RNNs are the current standard architecture for communication tasks like H2 (Karpowicz et al., 2024; Willett et al., 2021). Training and tuning was less stable than for our continuous decoding tasks and required more extensive hyperparameter tuning, perhaps because the overall dataset size remains small (<1k samples), specific parameters are listed in the codebase. We observe three regimes in both training and fine-tuning. First, the model can fail to achieve an initial learning period. Second, the model can achieve reasonable nontrivial solutions, comparable to expected performance for unaugmented RNNs (though we do not quantify this). Third, some models will exhibit learning instabilities that resolve in significantly improved performance. We illustrate these regimes in example validation curves below. Overall, the third regime is rarely achieved. More relevant to the main narrative of this work, fine-tuning appears to degrade both final solution quality and reduces the range of nontrivial hyperparameters (not shown). Investigating a sequence to sequence objective over CTC loss would be valuable future work.

### B.5  MULTISCALE DECODING ON INDIVIDUAL MOTOR TASKS

Fig. 11A plots model performance for each of the 31 evaluation settings we study in the eight primary evaluation datasets we use. Studying any individual dataset will yield variable conclusions on whether pretraining structure is helpful, underscoring the need for proposed foundation models to be evaluated across many different datasets. Here specifically we see the most clear scaling with pretraining data (color gradient with red on top) in the Critical Stability Task and Bimanual Task. FALCON tasks and Self-paced Reach appear minimally affected by scaling pretraining data, in that

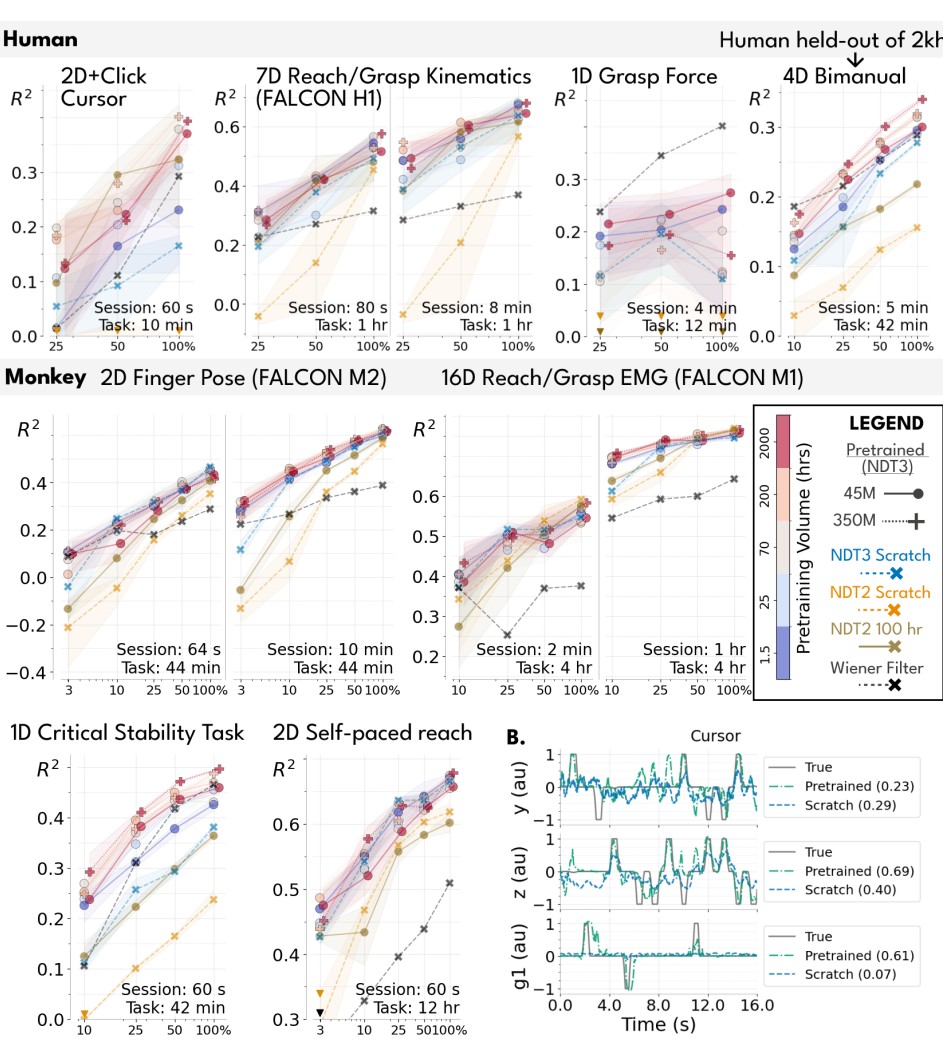

**Figure 11.** A. Fine-tuning evaluations for individual datasets. Performance on both held-out (left) and held-in (right) splits are shown side by side by FALCON datasets. We shade the standard deviation of 3 model seeds in fine-tuning. Different tasks show substantial variability in benefit from pretraining. B. We show example predictions of a pretrained (45M 200h) and from-scratch NDT3 for the 2D + Click Cursor task to give a sense of what different prediction performances mean in terms of open loop data prediction. Numbers in legend are the $R^2$ for that model's predictions in the shown snippet.

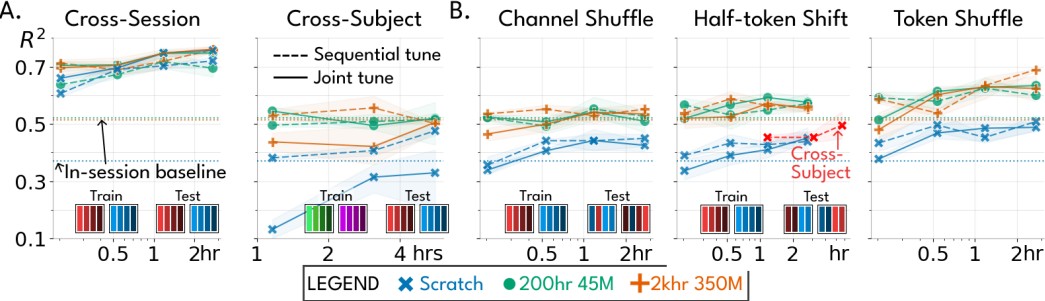

**Figure 12.** A replication of Fig. 4A/B, but showing results for both sequential and joint fine-tuning in each setting. As before, each model here tunes with some data from other settings (e.g. cross-subject data for the cross-subject panel), and a fixed amount of data on the test session. Sequential tuning first tunes on other-setting data, and then tunes on test session data. Joint tuning uses all data at once. In Fig. 4A/B, we only showed the better choice for each panel, i.e. we showed joint tuning for cross-session data, and sequential tuning for the other settings. In addition, we overlay how the cross-subject tuned models perform when applied to half-token shifted test data, confirming that subject and session data transfer similarly to shifted test data.

either pretrained models are generally slightly above a from scratch model at all data scales with no particular best pretrained model. The 2D + Click and Grasp datasets uniquely show high variability in model performance and strong degradation of the 350M 2 khr model at low data scales. Grasp instability was so high that we trained 9 seeds instead of the standard 3 to better estimate model performance. We propose this degradation is due to the instability of full fine-tuning of large models at the extremely low data scales these datasets present (e.g. 2.5 minutes at the 25% scaling). Finally, we remind that the 2D + Click, FALCON H1, and 1D Grasp Force tasks are datasets from human participants that are included in the 2 khr pretraining. Surprisingly, we see no particular benefit to the 2 khr model.

These scaling plots also provide more precise context for baseline performance. NDT2 performs particularly poorly in the low data regime, while Wiener Filters perform poorly in the high data regimes.

In Fig. 11B, we illustrate qualitative predictions on private datasets. These visualizations show a diversity in covariate timescales and structure. They also illustrate that the summary $R^2$ obscure several features of model predictions. For example, pretrained models in Cursor Y tend have false positive deflections in movement. $R^2$ also is not easily comparable in tasks with continuous dynamics (CST) vs. transient dynamics (Cursor G1).

## B.6 SEQUENTIAL TUNING IS SIMILAR TO JOINT TUNING

In Section 3.2, we showed that channel shuffling and half-token shifts were sufficient to reduce cross-session transfer to the extent of cross-subject transfer. Here we add a methodological subtlety on how the tuning is done. Given cross-context data and a test session, we can either jointly tune on all data (as we do in our primary evaluation), or sequentially tune on the cross-context data and test session. We find sequential tuning is particularly necessary for successful subject transfer of from-scratch NDT3 models, but that it slightly underperformed joint tuning on cross-session models (Fig. 12A). Seeing that sequential tuning is mainly advantageous for from-scratch models, we speculate that sequential tuning is particularly helpful for filtering learning signals in highly heterogeneous data. In Fig. 4A/B, for clarity, we reported jointly tuned results for cross-session data and sequentially tuned results otherwise.

## B.7 AGGREGATE PERFORMANCE ON ALL NDT3 MODELS WITH SIGNIFICANCE TESTS

We additionally report the average performance of an NDT2 model pretrained with 100 hours of human data and two NDT3 models pretrained with 25 hours and 70 hours. These models are placed in context with the models from Fig. 3D, in Fig. 13A. Note that for NDT3 models each successively larger data scale uses a strict superset of data from smaller scales. We also provide the p-values computed for the significance of the difference between each pair of models in Fig. 13B. P-values are computed as FDR-corrected pairwise t-tests. The 350M 2 khr model has $p < 0.06$ improvements

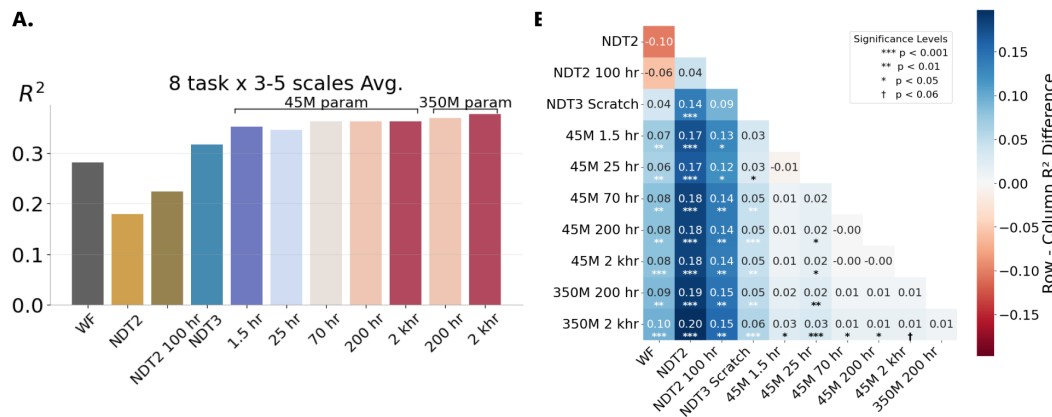

**Figure 13.** A. A replication of Fig. 3D including an additional 25 hr and 70 hr model. The two additional models show the precise performance we measure may be noisily related to to pretraining data scale. B. Heatmap of differences between performances of pairs of models with significance tested with FDR-corrected pairwise t-tests. Note that coloring is used here to indicate differences, not significance. Positive numbers with significance indicate the row model outperforms the column model.

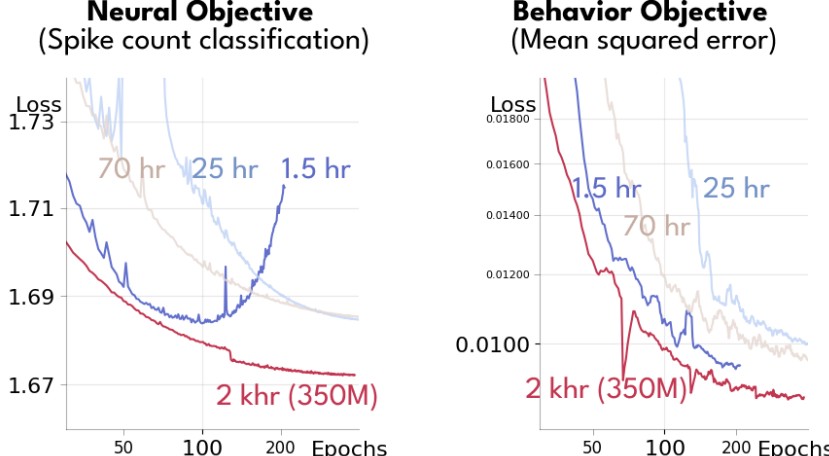

**Figure 14.** Pretraining curves shown for 45M parameter NDT3s at 1.5 hour, 25 hour, and 70 hours, in addition to the curves for the 2 khr 350M parameter NDT3. We separately show metrics on neural and behavioral objectives through training. Since early stopping is used in model selection, we verify here that neither objective is overfits significantly, except for the 1.5 hour model's neural objective.

over all but the 350M 200 hr model. Interestingly, all other pretrained models, except the 25 hr model, appear equivalent, at least statistically. We presume this is due to the fact that our evaluation of 31 task settings may be insufficiently large.

NDT2 performs relatively poorly in our evaluation. This is true even when tuning from the public checkpoint trained on 100 hours of neural data from humans, though tuning does in general improve over the from-scratch NDT2 training. We believe pretrained NDT2's performance gap with NDT3 is partially due to NDT2's need to newly initialize decoding layers in each downstream task, which increases NDT2's dependence on thorough hyperparameter tuning. This makes NDT2 a poor candidate for a foundation model. Section C.3 and Section C.6 describe a number of methodological innovations that likely each contribute to the remaining performance differences between NDT2 and NDT3.

### B.8 NEURAL VS BEHAVIORAL OBJECTIVES

In this work, the neural objective is present mainly as an auxiliary objective to improve downstream decoding. We see that neural and behavioral objectives are complementary in Section B.3.

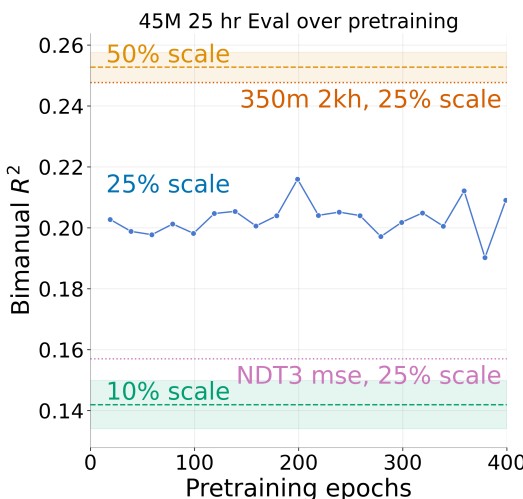

**Figure 15.** Downstream evaluation conducted through pretraining of a 45M 25 hour NDT3, for the 25% scaling of the bimanual task, provided in context of the final evaluation achieved by the 10% and 50% bimanual task scale settings for the 45M 25 hour models, and the 25% scalings for the from-scratch NDT3 and best NDT3 (350M 2 khr). The benefits from pretraining are attained by epoch 20, and plateaus without overfitting for the rest of pretraining.

In Fig. 14, we provide some additional context, showing that neural objectives also improve through pretraining, i.e. that we are not overfitting the neural objective and thus degrading decoding.

### B.9 DOWNSTREAM PROBE THROUGH PRETRAINING

Upstream and downstream performance are generally assumed to be correlated in pretraining studies. This assumption supports the use of a single evaluation at the end of pretraining, rather than throughout pretraining, particularly if pretraining is not overfit. It is possible, however, that the narrower tasks used in evaluation may be learned or even overfit earlier than the general pretraining task. To assess this, we conduct a small downstream probe of checkpoints every 20 epochs of pretraining for a 45M 25 hour NDT3. This probe measures the performance specifically at the 25% scaling of the bimanual task. In Fig. 15, we show this progression and compare the variability of checkpoints in pretraining with the differences across different task scales and pretraining models. The downstream probe for this task shows quick benefit from pretraining, providing the full gain over from-scratch models by the first checkpoint we evaluate at epoch 20, and then plateaus. This contrasts with the pretraining plots in Fig. 14. The mismatch between upstream and downstream progress here differs from correlated upstream-downstream progress in single-epoch language model studies (Yang et al., 2024), which may be due to differences in domain or pretraining scale.

### B.10 EVALUATION ON THE NEURAL LATENTS BENCHMARK

The Neural Latents Benchmark (Pei et al., 2021) is a benchmark for evaluating latent variable modeling of neural activity. This evaluation differs from NDT3's direct decoding evaluations in that all of NLB's metrics are derived from acausally inferred neural firing rates, akin to a pixel-level objective in computer vision. Note specifically that the decoding metric officially reported in the NLB is derived from submitted firing rates on the evaluation server; report decoding performance on NLB datasets without modeling neural activity is not an expected use of the benchmark.

Despite its distinct setting, the NLB provides two well-established datasets on which to evaluate models of motor cortical activity, providing context for NDT3's performance outside of decoding. To apply NDT3 to the NLB, we tuned NDT3 in a new downstream task where insert new a token at each timestep, from which we linearly decode firing rates of held out neurons. We performed this tuning separately for each of the maze and random target tasks (RTT), reporting the resulting co-bps scores in Table 1. NDT3 performs poorly. From-scratch training on the single-session benchmark datasets underperform NDT1 in all tasks. Tuning from pretrained models improved performance,

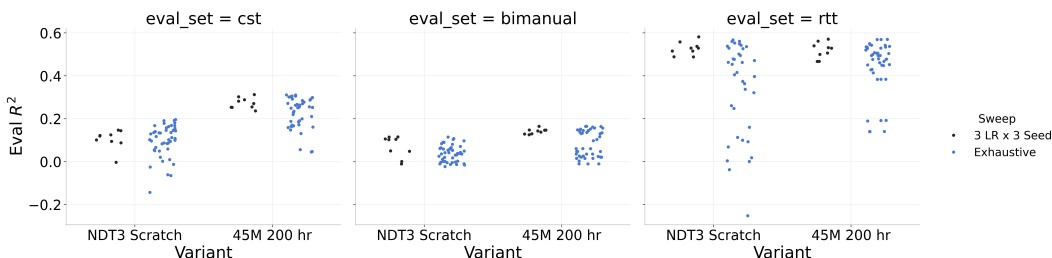

**Figure 16.** For 3 monkeys datasets at 10% scale, we extend a HP sweep to 5 LRs and dropout in $[0.0, 0.1, 0.3]$ (vs default 0.1). For fine-tuning, we also sweep weight decay in $[0.0001, 0.01, 0.1]$ (vs default 0.1), while for from-scratch models we also sweep Transformer width ($[256, 512, 1024]$) vs default 512. This yields a 45-model sweep on 1 seed. We compare the range of scores achieved by this larger sweep against the standard 3 LR x 3 seed sweep.

sample efficiency, and robustness to hyperparameters (only performance is reported here), but did not dramatically change model competitiveness. Zhang et al. (2024) reported the value of more diverse neuron-level objectives for neural activity prediction, though their submission also dramatically underperforms NLB SoTA, warranting critical examination of whether pretraining benefits low-level modeling of neural activity.

| Dataset | RTT | Maze | Maze Large | Maze Medium | Maze Small |
|---|---|---|---|---|---|
| NDT1 | 0.1643 | 0.3597 | 0.3739 | 0.3081 | 0.2788 |
| SoTA | 0.2010 | 0.3650 | 0.3831 | 0.3329 | 0.3458 |
| NDT3 Scratch | 0.1533 | 0.3093 | 0.2892 | 0.1859 | 0.1853 |
| NDT3 350M 2kh | 0.1695 | 0.2775 | 0.2781 | 0.1937 | 0.2116 |

**Table 1.** NDT1 vs NDT3 co-bps (higher is better) on NLB's MC Maze and MC RTT datasets in the 20ms split. The NDT3 results were produced were the standard sweeps used in this work, e.g. three random learning rates and three random seeds. SoTA results come from LangevinFlow for RTT and Maze and MINT for the Maze scaling datasets.

## C METHODS

### C.1 METRICS AND EVALUATION

Throughout this work we evaluate offline decoding of continuous covariates timeseries. The metric we specifically use is the coefficient of determination, $R^2$, as computed by scikit-learn's *r2_score* function. $R^2$ is a useful metric over MSE as 1 represents perfect prediction and 0 is the score achieved by best-guess baseline, the mean of the data. In pretraining, $R^2$ is computed over the flat average of all covariate dimensions, since each datapoint has differing covariate dimensionalities. In evaluation, $R^2$ is computed as a variance-weighted average of $R^2$s in each covariate dimension. Another difference between training and evaluation metrics is that training predictions are made over batched data, while evaluation predictions are mostly computed in a *streaming* fashion. Streaming requires continuous neural data across different behavioral epochs, and so cannot be performed for the Oculomotor and CST datasets. We also omit it for the motor cortex self-paced reach dataset, which has a very large evaluation split. Streaming allows timesteps at the beginning of each sequence to leverage neural context from the preceding sequence, which raises performance slightly, as shown in the continuous vs trialized analysis (Section 3.3). We limit history in streaming evaluations to the max history seen in tuning (1 second).

### C.2 TRAINING

Pretraining hyperparameters were manually tuned in preliminary experiments at the 45M parameter models on small datasets. 350M models diverged at the chosen $4e-4$ peak LR, so we lowered peak LR to $1e-4$. For tuning, the explored LRs are $1e-4, 3e-4, 5e-4$ for training from scratch and $3e-5, 1e-4, 4e-4$ for fine-tuning. While this is far from an exhaustive search, we show in Fig. 16 that other regularization hyperparameters are set to reasonable defaults such that this

sweep finds near optimal results for both a from scratch model and fine-tuning the 45M model. Fine-tuning, like pretraining, is early stopped with a patience of 100 epochs. Batch size is uniformly set to 16K in pretraining, and scaled to be roughly 10-20% of dataset size in fine-tuning. NDT3 from-scratch models were trained at the 11M parameter range. Exact model configurations for different experiments are documented in the codebase.

NDT3's simple architectural design allows us to train on batches from different tasks and dimensionalities. To avoid excess padding in training, we concatenate pretraining data that is otherwise discontinuous (trialized) into 2 second data. We do not add any separator tokens, as this does not appear to have a performance impact for language models (Geiping and Goldstein, 2022). With mixed-precision training, the 350M parameter NDT3 can fit the 4-8K tokens in each input context in the memory of 40G NVIDIA A100 GPUs. Thus we can restrict NDT3's pretraining parallelism to data-parallelism.

Using Kaplan et al. (2020)'s equation for FLOP computation, $C_{\text{forward}} = 2N + 2n_{\text{layer}}n_{\text{ctx}}d_{\text{attn}}$, we compute the footprint of the 350M 2kh model. We use about 0.9B FLOPs per token in the forward pass, and about 0.9T neural tokens processed over training, which yields a pretraining footprint of about 2.4e21 FLOPs.

## C.3 BASELINES

**Wiener Filter** The Wiener Filter baseline was cross-validated over regularization strength. We also swept history of neural input up to the max length provided to NDT, and reported the $R^2$ of the best WF according to test data in primary evaluation (slightly advantaging the WF). Generalization plots in Section 3.3 report the performance of WF models at these different histories. For evaluating angular generalization, WFs were only swept up to 1s history due to memory limits; performance was not varying substantially with history so we do not expect this to have impacted conclusions. The WF was for simplicity directly fit on the concatenated trial data, which may have slightly negatively impacted its performance in trialized datasets (Oculomotor, CST, Generalization analyses).

In the primary evaluations in Section 3.1, we considered WFs fit either independently per session in a dataset or jointly on all sessions, which is helpful for sessions in very low data regimes. We report the better of the 2. In generalization analyses, for simplicity, we only report joint fits, which may cause a slight downward bias in performance.

| Dataset | Patience | Held-In $R^2$ | Held-Out $R^2$ |
|---|---|---|---|
| H1 | 100 | $0.567_{\pm 0.034}$ | $0.453_{\pm 0.030}$ |
| H1 (reproduction) | 250 | $0.628_{\pm 0.011}$ | $0.517_{\pm 0.016}$ |
| H1 ((Karpowicz et al., 2024)) | 250 | 0.62 | 0.52 |
| M2 | 100 | $0.563_{\pm 0.015}$ | $0.352_{\pm 0.028}$ |
| M2 (reproduction) | 250 | $0.582_{\pm 0.002}$ | $0.391_{\pm 0.009}$ |
| M2 ((Karpowicz et al., 2024)) | 250 | 0.63 | 0.43 |

**Table 2.** NDT2 H1 and M2 results when trained with 100 epochs of patience (this work) in fine-tuning vs 250 as in Karpowicz et al. (2024). We report mean and standard deviation of 3 model seeds on the FALCON evaluation (which is in turn a cross-session mean).

**NDT2** NDT2 baselines were prepared with its public codebase. We trained NDT2 models both from-scratch and from the public checkpoint pretrained on 100 hours of human data. Max context length and patience were held constant across the models. This restriction to a patience of 100 epochs accounts for some difference with the reported FALCON benchmark results in Karpowicz et al. (2024), as we note in Table 2. Other choices were left to NDT2 defaults. For example, NDT2 uses z-score normalization, which we kept. In from-scratch training, for simplicity, we jointly trained NDT2 with its neural reconstruction loss (masking of 25%) and a supervised decoding loss. This is true for all eight evaluation tasks except CST. In the CST task, we used only the supervised decoding loss, as the token dropout used in reconstruction can dropout all neural input.

For hyperparameter tuning, we matched NDT3's tuning budget for the pretrained NDT2 checkpoint by only exploring 3 learning rates. Given mediocre NDT2 from-scratch performance, we swept NDT2 over 2 model sizes in addition to the standard 3 learning rates. We set the NDT2 from-scratch

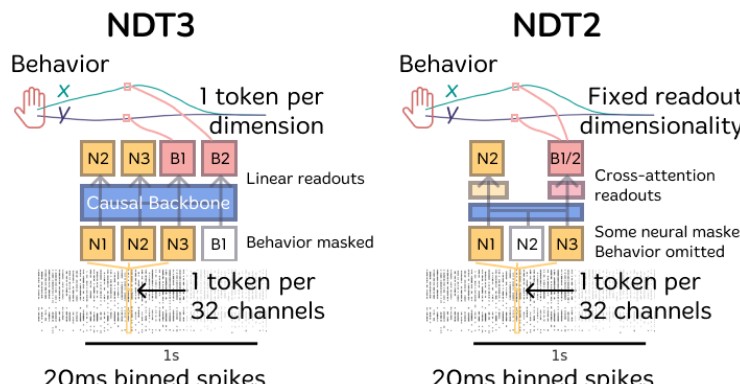

**Figure 17.** NDT2 and NDT3 architectures differ mainly in the conversion of NDT2's representation learning backbone to NDT3's single multimodal stream. NDT3 directly intakes neural and behavioral tokens and predicts with a next-step objective. NDT2 employs explicit masking of input neural tokens and extracts neural and behavioral predictions at each timestep with cross-attention layers.

model sizes to 20M and 72M to be comparable with NDT3 45M, but note that the NDT2 pretrained checkpoint is only 6M parameters.

**NDT2 vs NDT3.** NDT3 builds off of NDT2 but departs in several manners to enable more streamlined scaling and analysis over heterogeneous decoding tasks, which we overview in Fig. 17. Lower level technical changes are described in Section C.6. Both models relate neural data to behavior and train with both neural data and behavior prediction objectives. Both models use both these objectives in fine-tuning, but NDT2 only uses the neural objective in pretraining. NDT2's neural data objective is based on MAE (He et al., 2021), such that some fraction of input neural data tokens are masked and reconstructed in a decoder separate from the main backbone. However, this explicit masking was originally developed to study representation learning on images, not timeseries decoding. Causal domains like BCI control can also learn nontrivial representations simply through next-step prediction. NDT2 employed both explicit MAE masking and a causal attention mask in its backbone, which is redundant computationally and also reduces the context available to make predictions. NDT3 thus dispenses with the masking mechanism and uses next-step prediction alone. NDT2 also differs from NDT3 in its readout of behavior. Again, since NDT2 studied representation learning, it used additional cross-attention readout layers to "probe" behavior predictions at each timestep. NDT3 simplifies this two-part encoder-decoder design to a decoder-only design. In this flow, masked behavior tokens are provided at the input and filled in through the backbone, and varying the input tokens allows us to make predictions of the corresponding behavior dimensionality.

### C.4 PRETRAINING AND EVALUATION DATASETS

Pretraining datasets were comprised of historical data from several labs, the rough composition of which is shown in Fig. 2B. The evaluation behavior used during pretraining was reaching in 2 monkeys. The first monkey dataset came from a public release (Flint et al., 2012), and the second from a private dataset ( REDACT  lab). The latter had center-out reach in standard conditions and under visual feedback perturbations. The monkey in the second dataset is also present in the 1khr monkey and 2kh and up model dataset sizes, though performing in a different set of experiments.

Inherent to the process of large-scale scraping is a loss of detail on what precise tasks were used, so we only have a qualitative description of tasks we believe are well represented. NDT3 trains on a wide variety of reaching behaviors from relatively constrained (2D center-out reaching to fixed number of targets) to relatively unconstrained (self-paced, more targets, potentially 3D) and under experimental manipulations (delayed onset, multiple targets, different error thresholds requiring more precision). These reaching behaviors are described in both endpoint kinematics and as EMG. A smaller fraction of pretraining data are isometric and force related (force exerted against manipulandums) for wrist and arm motion. Human datasets contain a variety of iBCI tasks, with closed loop datasets reflecting both high and low quality control. These tasks include reach and grasp behavior from 1-10 degrees of freedom, as well as some individuated finger tasks for clicking.

We detail evaluation datasets in Table 3. Three datasets come from the FALCON benchmark (Karpowicz et al., 2024), two are based on public datasets ((O'Doherty et al., 2017; Deo et al., 2024)), and three are private. Note we avoid the Neural Latents Benchmark (Pei et al., 2021) as it does not directly measure decoding performance. For each evaluation dataset, we specify a tuning split and an evaluation split. Only tuning split data is changed when varying data scale. Tuning and evaluation splits are block-contiguous, i.e. trials are not interleaved, for better downstream applicability.

## C.5 GENERALIZATION ANALYSES AND FURTHER EVALUATIONS

**Intra-session generalization** Posture, spring, and angular generalization evaluate OOD performance in the standard setup of comparing in-distribution and out-of-distribution performance directly (with changes in the underlying evaluation dataset) The intra-session temporal shift analysis is evaluated in an inverted, slightly more rigorous setting. Specifically, we trained two sets of models on the two different temporal blocks, and evaluated on an evaluation split in the later block, rather than only training on the early block and evaluating on both blocks. This way, the OOD shift is measured with respect to the same evaluation dataset.

## C.6 ARCHITECTURAL DETAILS

NDT3 adopts several architectural innovations used in recent Transformer models. These were compared against baselines in preliminary experiments, but formal ablations in the final experimental setting were not conducted. We defer full description of the Transformer dimensions to the public codebase.

- FlashAttention 2 (Dao, 2023) is used to increase training and inference speeds. On the NERSC Perlmutter cluster, with FA2, 45M NDT3 trained at about 270M neural tokens per 40G A100 hour, 350M NDT3 trained at about 70M neural tokens per A100 hour. Note, FA2 also enables use of the 350M model for real-time (<20ms) inference latency.

- Positional Embeddings (Su et al., 2023): Rotary embeddings are applied to indicate the real-world timestep of every input token. Additionally, 48 categorical learned embeddings are reserved to distinguish token modality and position within a timestep (10 for neural, 16 for covariates, 16 for covariate constraints, 1 for reward/return, 1 for dummy tokens, remainder unused).

- QK Normalization (Dehghani et al., 2023; Wortsman et al., 2024): An additional layer norm is applied to the query and key embeddings, before the rotary embeddings, which helped stabilize training of the 350M parameter models.

- No context embeddings (Ye et al., 2023): Differing from NDT2, no learned embeddings for disambiguating input datasets were prepended to each input. This was removed for simplicity. Per GATO (Reed et al., 2022) and language modeling practices, we instead leave task / dataset disambiguation to the modeling process: In pretraining, the covariate maskout strategy allows for many tasks to be specified in-context (as later behavior can be inferred on the basis of earlier neural-behavioral token relationships). In fine-tuning, the tuning dataset already uniquely specifies the function to be learned.

- Cross entropy loss for spiking data prediction: We used the standard cross entropy loss to classify spike count over the Poisson loss common in many neural data architectures. Since the overall ablation of neural objective shows no large impact in this work, it is likely that this decision should be evaluated with neural data related tasks rather than decoding.

We document the Transformer model shapes considered in our work in Table 4. This shape is not systematically explored in our work, and is by historical artifact, slightly different than the shapes used in NLP/CV. Embedding parameters are negligible. One possible area of interest is that the feedforward expansion factor is 1 in our model, i.e. the MLP dimension is low. If MLPs do serve as memory stores in Transformers (Geva et al., 2021), increasing this shape may yield more performant model size scaling, given the heterogeneity of our datasets.

# D NDT3 MODEL CARD

The card is currently only provided in the codebase.

**Table 3.** Evaluation datasets used for multiscale decoding and generalization analyses. The references provide extended description of the behavioral task. Dashed line separates datasets for Section 3.1 and for analysis. Datasets use unsorted multi-unit activity and are processed in 1s chops unless otherwise mentioned.

| Dataset | Description |
|---|---|
| FALCON H1, M1, M2 (Karpowicz et al., 2024) | 3 separate single-subject multi-session datasets for different iBCI tasks. Data comes in a high data split (held-in), and a low-data split (held-out), with the intention on identifying methods that can achieve parity in the two settings. H1 is an open loop human dataset for calibrating 7D reach-and-grasp in a robot arm. M1 is a monkey reach-and-grasp task to different objects with EMG recordings. M2 is a monkey 2D finger movement task with manipulandum-measured kinematics. Scaling scores are reported on the test set. |
| Self-paced reach (RTT) (O'Doherty et al., 2017) | Monkeys reach for random targets one at a time in a small planar workspace. We decode 2D arm velocity in monkey Indy. Has neural data from M1 and S1, we use M1 in Section 3.1 and Section 3.2 and S1 in Section 3.3. |
| Bimanual Cursor Control (Deo et al., 2024) | A human open loop dataset where the participant attempts movement of one or both hands to control two cursors. |
| 2D Cursor + Click (private) | Cursor control is a classic iBCI endpoint (Pandarinath et al., 2017; Wolpaw et al., 2002; Jarosiewicz et al., 2015). Two human participants attempt movement according to visually cued cursor movement and audiovisual click cues. We also use this dataset for trial structure analysis in Section 3.3. |
| Grasp force (private) | A open-loop dataset with two human participants attempting isometric power grasps. Specifically, participants were asked to match force output according to visual cues in a Mujoco environment. Grasps cued were both static (instant onset, hold, and offset) or dynamic (gradually increasing force). This dataset is valuable for human iBCI study because force modulation is required in many motor behaviors, and grasp force has primarily only been characterized in monkeys until now (Branco et al., 2019). Uses 2 second intervals due to long behavior timescale. We expect this dataset can be released by end of 2024. |
| Critical Stability Task (Quick et al., 2018) (private, trialized, sorted) | A monkey dataset collected to study continuous control relative to ballistic movement. The monkey balances a virtual cursor on a 1D workspace for up to 6 seconds. |
| Posture-varied Center-Out (Marino et al., 2024) (private, trialized, sorted) | A monkey center-out task, but the monkey's hand is adjusted to one of 6 different starting positions. We use the central position as center and the rest as edge. |
| Spring-load (Mender et al., 2023) | A monkey moves fingers, clamped together in a manipulandum for effective 1DoF, is neutral or under spring load. |
| Center-out, Monkey J (Ma et al., 2022) (trialized) | Used in Section 3.2. A monkey performs an isometric center out task. Forces are measured by the manipulandum and converted to cursor velocity signals. |
| Center-out, Monkey V (private, trialized) | Used in Section 3.2. A monkey reaches to one of 8 radially arranged targets by moving a manipulandum (Kinarm). |
| Oculomotor pursuit (Noneman and Patrick Mayo, 2024) (private, trialized, sorted) | A monkey visually tracks (via smooth pursuit) a target that moves from center of workspace to one of four directions. A few dozen neurons are recorded on probes in each of frontal eye field (FEF) and area MT. We decode pupil velocity. The small number of neurons in this dataset required resetting NDT3 neural readin/readout layers. |
| FALCON H2 | Human open loop dataset where a participant attempts movement to write letters cued on a screen (Willett et al., 2021; Fan et al., 2023). The large number of timesteps in this dataset required resetting NDT3 neural readin/readout layers (to use fewer neural tokens). |

| Model | Layers | Width | MLP Size | Heads | Parameters (M) |
|---|---|---|---|---|---|
| NDT2 PT (Ye et al., 2023) | 4 | 256 | 256 | 4 | 6 |
| NDT3 Base | 6 | 1024 | 1024 | 8 | 45 |
| NDT3 Big | 12 | 2048 | 2048 | 16 | 350 |

**Table 4.** Transformer Model Shapes used in this work.

