# OpenReview forum: "A Generalist Intracortical Motor Decoder"
_ICLR.cc/2025/Conference — Submitted to ICLR 2025_

### Official Review · Reviewer_Kgow · 2024-10-18

**Soundness:** 3
**Presentation:** 3
**Contribution:** 3
**Rating:** 6
**Confidence:** 3

**Summary:**

The authors developed Neural Data Transformer 3 (NDT3), a neuroscience foundation model for decoding movements. They pretrained NDT3 using 2,000 hours of neural data, paired with various motor target labels, from over 30 monkeys and humans across 10 labs. Following this, they demonstrated the limitations of the pretrained foundation model on certain downstream datasets.

**Strengths:**

This is the **largest dataset** used so far. A highly relevant previous study (Azabou et al., NeurIPS 2023) utilized 100 hours of recordings from 7 monkeys, whereas this paper includes 2,000 hours of recordings from 30 monkeys and humans. With this unprecedentedly large dataset, the authors found that scaling from 200 to 2,000 hours of data necessitates increasing the model size from 45M to 350M.

**Weaknesses:**

**Contribution of NDT3:**\
The primary contribution I gather from this paper is the limitation of scaling when dataset size increases. To my knowledge, there are at least six foundation models in the neuroscience field (listed by year):

NDT, Ye and Pandarinath, NBDT 2021\
**EIT**, Liu, ..., Dyer, NeurIPS 2022 (monkey motor tasks)\
NDT2, Ye, Collinger, Wehbe, Gaunt, NeurIPS 2023\
**POYO**, Azabou, ..., Dyer, NeurIPS 2023 (monkey motor tasks)\
Wang, ..., Tolias, bioRxiv 2023/2024 (mouse visual cortices)\
Zhang, ..., Dyer, Paninski, Hurwitz, arXiv 2024 (mouse brain regions)

NDT3 has only been benchmarked against NDT2 and the Wiener Filter (WF) but not against two highly relevant baselines: **EIT and POYO**.

**Writing Quality:**\
The writing in this paper is poor, making it challenging to read due to some typos in the text.

Examples when referring to Figures:\
L107: There is no left or right panel in Fig. 2A; it should be Fig. 2B.\
L466: It should be Fig. 5B instead of Fig. 5C.\
L479: There is no Fig. 5D; it should be Fig. 5C.

A difficult-to-understand sentence:\
L214-215: "Other Transformer variants have been proposed for pretraining on spiking data (Azabou et al., 2024; Zhang et al., 2024), but the field yet lacks consensus benchmarks to distinguish the most promising proposal to scale."
Does “benchmark” here imply that the authors plan to benchmark their Causal Transformer against other Transformer architectures? In this study, they only benchmarked NDT2, which used the MAE structure. However, is the performance difference due to the different structures or different data sizes?

**Minor Issues:**\
L200: Extra word: Prior work (Azabou et al., 2024; Ye et al., 2023; Zhang et al., 2024) **ran** focused evaluations by tuning...\
L277-278: Missing parentheses: ...that increasing model size and dataset size in tandem is important for performance **(**Dosovitskiy et al., 2021; Kolesnikov et al. 2020; Aghajanyan et al. 2023**)**\
L392: Remove the extra parentheses: ...which fail completely **(**(Rizzoglio et al., 2022)**)**.\
L482: Missing a word: While the former can be attributed **to** the close interaction of sensorimotor areas...

**Questions:**

Why was the MAE Transformer structure in NDT2 replaced with the Causal Transformer in NDT3?

Why are the red lines broken in Fig. 3A and 3B? Does this indicate that R² suddenly dropped between 100 and 200 epochs?

Regarding the dimensionality of movement covariates (Fig. 2A), why could it be as high as 10? My understanding is that if a hand is moving on the XY plane, the dimensionality would be 2. Can a hand move in 10 dimensions?

---

> ### Author Response · Authors · 2024-11-14
> **Respone to kGOW**
>
> _Overall_: We thank the reviewer for their detailed feedback (and catching of several typos). The two main critiques were the insufficiency of baselines and the poor writing quality. In the detailed response we comment on why it is difficult to compare with the baselines you have requested, but we are beginning experiments comparing with the public pretrained NDT2 model, which we hope addresses the concern on baselines. As for the poor writing, we have amended the typos you have pointed out, and we are happy to receive further feedback on any low and high level area where the reviewer finds the writing confusing so that we might improve the text and overcome this issue of writing quality.
>
> _Weakness - Contribution / Baselining_: We would like to clarify the relationship of the listed models to NDT3. NDT and EIT are single-session models, trained on a single day’s data, and are not foundation models. EIT further does not provide a codebase. POYO introduces a pretraining method and trains a model with up to 100-200 hours of data, but POYO does not release pretrained weights, which makes it difficult to compare against their method without greatly increasing the scope of our work (evaluating scaling on a fairly distinct architecture). Given the challenges to comparing against other works in our own paper, we have reported NDT3 on the newly released FALCON benchmark makes it easy to compare against NDT3 in future works. See also our global response on baselining.
>
> _Weakness - Writing Quality_: We apologize for the typos in the text and have corrected your examples. For L214-215, we are referring to benchmarks, the noun, not the action of evaluating different models. We are saying that there is no third party dataset with fixed preprocessing and private evaluation that many people in the field submit to for decoding, like ImageNet. The NLB benchmark is referenced in POYO, for example, but we believe its use is inappropriate (see response to USSQ for details). We are not claiming that NDT3’s architecture is superior to NDT2 in general, only that it is easier to scale to decoding over large amounts of heterogeneous data for pretraining. However, the fact that from scratch NDT3 achieves better decoding than from scratch NDT2 across our extensive evaluation does show that it may be easier to train high performance decoders with NDT3 than NDT2.
>
> _Question - Why Change NDT2?_: We changed the architecture from NDT2 to the causal transformer in NDT3 for two reasons. First, it allows us to use one model to decode data of different dimensionalities without needing different readout layers. A similar approach is used in other generalist models like [TDM](https://arxiv.org/abs/2305.10912). Second, FlashAttention2, which allows several-times faster pretraining, does not support unconventional masking strategies. The open source implementation of NDT2 relies on a shuffling strategy that requires such an unconventional attention mask and is not easily compatible with FlashAttention 2’s requirements. We have added text to address this question in C.3.
>
> _Question - Why broken lines?_: Yes, this is attributed to an explosion in loss. This occurs in several of our pretrained models but is also not uncommon in pretraining overall (we believe), and generally is not known to degrade the quality of the outcome models at convergence.
>
> _Question - Dimensionality > 2_: NDT3 is pretrained on data from many different behavioral tasks. While some tasks are 2D, as you mention, other tasks, like when decoding recordings from different muscles in EMG, have many more target dimensions to decode. You can see these examples in Fig 2B. There are also high-dimensional kinematics, as in FALCON H1, which is 7D robotic upper limb control.
>
> We hope these answers address some confusions and again, welcome further critiques on writing quality in the new section on experiments.

---

> > ### Comment · Reviewer_Kgow · 2024-11-25
> > **Releasing pretrained weights**
> >
> > I appreciate the authors' responses to my questions.
> >
> > The presentation looks good right now.
> >
> > I just checked the GitHub page of POYO, and they indeed have not released their pretrained weights. There is even someone asking for the pretrained weights. However, the authors of the POYO paper promised: "*We will make the weights and code available, and provide both pretrained models as a resource to the community.*" For neuroscience foundation models, in my opinion, they are more like datasets or resource papers that people in the field can further explore in their own research. Most researchers have no access to those huge sets of training data and the expensive computational resources.
> >
> > Therefore, I would recommend the publication of this work if the authors could release their code and pretrained weights after acceptance.
> >
> > However, I am still confused by your explanation of the broken line as "attributed to an explosion in loss." What does this exactly mean? It creates a huge valley between 100–200 epochs. How can you ensure it will not explode again?

---

> > > ### Author Response · Authors · 2024-11-25
> > > **Weights are available (not shown here due to OpenReview anonymity)**
> > >
> > > Thanks to reviewer kGOW.
> > >
> > > We agree that the code / pretrained weights should be made available for these types of work. Our codebase is already attached in the supplement (and its release is ready, but a link is not provided here for anonymity), and has since been updated with a starter notebook for model fine-tuning and inference. Our weights are also available through Huggingface (again, not linked for anonymity).
> > >
> > > Regarding the broken line, if we plotted the loss instead of the R2 metric, there would be a transient spike in loss that decays back to where we'd expect the loss to be without the explosion. You can see these two views compared in Fig 8A/B. These losses are not very uncommon in large scale pretraining, though their exact nature is not understood. See discussion, for example, in [Wortsman 23](https://arxiv.org/abs/2309.14322) and [Cohen 22](https://arxiv.org/abs/2207.14484). NDT3 currently uses [QK normalization](https://arxiv.org/abs/2302.05442) to stabilize training, so the only model in the main text that actually experienced an explosion is the 200 hour model in Fig 3A/B. The 200 hour model in later evaluations is the unaffected "Other datasets" model.
> > >
> > > We are glad to hear you can recommend this work after our rebuttal, and would appreciate if the score could be changed once you are ready. Thanks again for your feedback.

---

> > > > ### Comment · Reviewer_Kgow · 2024-11-25
> > > > **Scores are updated**
> > > >
> > > > Since the authors have already (instead of promise) uploaded their pretrained weights, I increased all of my scores (Soundness, Presentation, Contribution, and Overall).

---

### Official Review · Reviewer_USSQ · 2024-11-01

**Soundness:** 3
**Presentation:** 3
**Contribution:** 2
**Rating:** 5
**Confidence:** 4

**Summary:**

This paper proposes large-scale pretraining of a Transformer-based foundation model, called NDT3, for motor decoding. Building on prior work by Ye et al. (2023), the model scales up to incorporate diverse datasets involving both monkey and human motor tasks.  The authors evaluate their pretrained model on eight downstream datasets and find that while some downstream tasks benefit from extensive pretraining, others show minimal improvement. They report that NDT3 model needs to be scaled up to 350M parameters to avoid performance drops when pretrained with 2000 hours of data. The authors also hypothesize that the poor generalizability may stem from the input-output shift, highlighting challenges in developing foundation models for neural data.

**Strengths:**

- Extending the scale of foundation models pretraining for neural population spiking activity. The authors have used very large amount of data (2000 hours) for pretraining a generalizable model for motor decoding. This represents a valuable step forward in investigating the effectiveness of foundation modeling approaches for motor decoding, and more broadly, for neuroscience applications.

- Building on prior research, the authors perform analyses to offer insights into scenarios where foundation models may be more effective. Understanding when and why these approaches are beneficial (or may fail) can illuminate domain-specific challenges for motor decoding.

**Weaknesses:**

- The main contribution of this work lies in extensive pretraining of large foundation models (45M or 350M parameters) using substantial amount of neural-behavioral data for better generalizability.  However, the performance gains from this heavy pretraining seems minimal and task-dependent. In several cases, the authors report that pretraining NDT3 with large datasets yields only minor improvements on downstream tasks. While the in-depth analysis of why generalizability has been limited is insightful (as outlined in the strengths), the results overall suggest minimal progress towards the goal. For example, in Fig. 3D, increasing the pretraining data volume from 200 hours (even from 1.5 hours) to 2000 hours only slightly improves the average downstream decoding.

- The authors find that to benefit from pretraining with up to 2000 hours of data, the model’s capacity needs to be significantly increased to 350M parameters. The goal of this extensive pretraining is to facilitate easy downstream use. However, finetuning such a large model entirely can itself be of significant cost. Therefore, it remains questionable whether or not scaling the model with data provides a practical solution.

- Lack of sufficient baselines: NDT3 has been primarily compared to itself under different data regimes or training from scratch. The authors additionally compare it to two baselines trained from scratch: 1) the predecessor model NDT2, 2) a very simple linear method (WF). These baselines seem insufficient for careful evaluation of NDT3. More expressive baselines such as MLP or RNNs should be added to strengthen the results. Additionally, without comparisons to other foundation models, it’s unclear whether the generalization challenges observed are unique to NDT3 or representative of foundation models for neural decoding more broadly. For example, comparisons with the foundation model in Azabou et al., 2023 will be needed, at least on the Neural Latent Benchmark. See next.

- The authors note that: “we avoid the Neural Latents Benchmark (Pei et al., 2021) as it does not directly measure decoding performance”.
I did not follow the above reasoning. Decoding is also assessed on this benchmark (NLB) it seems. Could the authors clarify how NLB differs from the evaluated datasets? It’s understandable that comparing to other foundation models may be challenging due to the lack of publicly available pretrained models. Nonetheless, aren’t there any other foundation models with available code and pretrained weights for comparison? For example, Azabou et al., 2023 evaluated their method (in terms of behavior decoding) on NLB datasets with motor tasks similar to those analyzed here. If benchmark comparisons on the authors’ data is not possible, NDT3 should be compared to such approaches including Azabou et al., 2023 on common downstream datasets such as NLB.

- Novelty compared with NDT2: Novelty compared with NDT2 seems limited and the model differences compared to NDT2 (if any) are not discussed in detail. It is not clear if NDT3 is the same model (potentially with more parameters) just pretrained with more data. In fact, the smaller NDT3 with 45M parameters seems to have comparable capacity to the NDT2 models used (20M and 72M parameters).  Please provide a detailed comparison of the model architectures, training procedures, and key innovations of NDT3 compared to NDT2. This would help clarify the extent of the novelty in NDT3. Also, related to the previous point, why is pretrained NDT2 not provided as a baseline in the main figures? Please provide this in the main figures. As NDT2 is also a foundation model and its pretrained weights are available, providing its finetuned downstream performance is a more informative baseline than the one trained from scratch e.g., in Fig. 3D and Fig. 10. Does pretrained NDT3 outperform pretrained NDT2? Please provide these in main figures and discuss.

- Novelty in terms of generalizability results: The utility of large-scale pretraining of foundation models and their transferability to downstream tasks for neural data (although not spiking activity) has been shown before. For example, Zhang et al., 2024 have proposed a foundation model for SEEG data with even larger pretraining data than NDT3. Similar to NDT3, Zhang et al., 2024 also evaluated the effect of model size on generalizability and showed improved performance with higher-capacity models. Therefore, the idea and conclusions about generalizability of foundation modeling with more brain data are not novel in the field.

References:

Joel Ye, Jennifer L Collinger, Leila Wehbe, and Robert Gaunt. Neural data transformer 2: Multi-context pretraining for neural spiking activity. In Thirty-seventh Conference on Neural Information Processing Systems, 2023. URL https://openreview.net/forum?id=CBBtMnlTGq.

Mehdi Azabou, Vinam Arora, Venkataramana Ganesh, Ximeng Mao, Santosh B Nachimuthu, Michael Jacob Mendelson, Blake Aaron Richards, Matthew G Perich, Guillaume Lajoie, and Eva L Dyer. A unified, scalable framework for neural population decoding. In Thirty-seventh Conference on Neural Information Processing Systems, 2023. URL https://openreview.net/forum?id=sw2Y0sirtM.

Daoze Zhang, Zhizhang Yuan, Yang Yang, Junru Chen, Jingjing Wang, and Yafeng Li. Brant: Foundation model for intracranial neural signal. Advances in Neural Information Processing Systems, 36, 2024.


Minor/questions:
- What datasets, pretraining, and analysis do the results in Fig. 1B correspond to? I could not find that information in the manuscript. Also, in this figure, it seems the pretrained model is falling short of scratch model when abundant task data is available. How should that be interpreted?

Errors:
- The main pdf file does not include the appendix. The complete document is provided as a supplementary pdf file.
- References are not listed in alphabetic order.
- Line 76: typo “motion”
- Line 86 parenthesis missing in “(Section 3.3)”
- Line 107: should be Fig. 2B not A
- Citations, Figs, etc don’t have links.
- Line 1272. Typo: “the”
- Line 1112: typo: “show”

**Questions:**

My questions are the ones stated in weaknesses and Minor/questions.

---

> ### Author Response · Authors · 2024-11-14
> **Response to USSQ**
>
> _Overall_: We appreciate the reviewer’s desire for rigor in the evaluation of NDT3. We agree that further baselining provides important context for the ultimate benefit of the NDT3 model, and will work to include baselines from the pretrained NDT2 checkpoint in this discussion phase. Considering that full-scale evaluations of new methods (even basic DNNs) are fairly expensive, we agree that external benchmarks are the natural place to compare additional methods, and will discuss these in detailed response. We also agree that the technical novelty and empirical benefit in this work is limited, but our contributions advance beyond model advances and also provide analyses showing where we can expect benefits currently and directions for future improvement (see global response to limited benefit).
>
> _Weakness - Limited benefit_: As mentioned, we agree with the assessment of limited benefit, which motivated our framing of the model as a generalist that performs many tasks well rather than setting records on every task. Generalists are similarly limited in other fields ([GATO](https://arxiv.org/abs/2205.06175), [RT-X](https://arxiv.org/abs/2310.08864). We also want to emphasize that limited gains from scaling are not unprecedented. For example, Fig 4. In [Entezari 21](https://arxiv.org/pdf/2302.13602) shows that several downstream tasks don’t benefit significantly from orders of magnitude more images in pretraining. Given this context, and given the extent of our other experiments, we hope to convey that limited performance is more an interesting challenge for the field rather than a failing of our methodology.
>
> _Weakness - Practical cost_: We agree that larger models are in general harder to use, but 350M parameter models are small for modern machine learning (BERT/GPT2 were this size in 2018) and represent the starting point for most works in foundation modeling. Their practicality can be demonstrated by the fact that consumer GPUs can tune models an order of magnitude larger, with the right tooling. While the necessity of scaling NDT3 specifically to 350M parameters is a concern, we suspect it is a concern for the field rather than a weakness of our method.
>
> _Weakness - Lack of baselines_: We chose to sacrifice baselines used so as to instead maximize the number of datasets we evaluated on (we already tuned over 2000 models). Nonetheless, we agree with the reviewer that baselines provide important context, even if only partially evaluated. To this end, at least for basic baselines, RNNs are already evaluated on the FALCON benchmark, where they uniformly underperform multi-session NDT2 models (see appendix of the [FALCON preprint](https://www.biorxiv.org/content/10.1101/2024.09.15.613126v1) and current [leaderboard](https://eval.ai/web/challenges/challenge-page/2319/leaderboard/5747)). We agree that the current scaling trends may not generalize to other foundation models, namely pretrained NDT2 or Azabou et al’s POYO. However, we believe that evaluating scaling trends on multiple candidate foundation models is out of scope, given that no model has been evaluated at large scale in this subfield. We are optimistic that our work can provide an important reference for future comparison, and have taken care to discuss the need for comparison with other methodologies in our discussion.
>
> _Weakness - NLB_: We believe this is an important nuance that has been confused both in the field. NLB is a representation learning benchmark, and decoding performance on the leaderboards is evaluated by remote training of a linear probe on submitted neural data firing rate predictions. In contrast, our evaluations directly predict the behavior, without explicit neural data modeling. Therefore it is somewhat inappropriate to evaluate NDT3 with NLB. In contrast the used FALCON benchmark directly evaluates behavior, though it is new and other models have not been tested on this benchmark yet.
>
> Further, we believe that Azabou et al. used the datasets from the NLB but preprocessed them for their own analysis and conducted private evaluation, which defeats the purpose of the benchmark. For example, there is no submission for POYO on the [NLB leaderboards](https://eval.ai/web/challenges/challenge-page/1256/leaderboard/3188), even though the reported score is almost 5% higher than other methods. Since we cannot compare to POYO's reports in the official benchmark, and there is no pretrained weight available for POYO, we would have to train and tune POYO ourselves to make a comparison based on architectures. Beyond POYO, we are only aware of pretrained NDT2 and [MtM (Zhang et al 2024)](https://ibl-mtm.github.io/)) for motor decoding. While MtM releases pretrained models, it is trained on mouse data outside the context of BCIs, and was released in July 24 (contemporary work).
>
> 11/24: We now report NLB evaluation in B.10, though as mentioned, these results are not comparable with POYO's reports.
>
> (Response continued in thread)

---

> ### Author Response · Authors · 2024-11-14
>
> _Weakness - Pretrained NDT2  1, Documentation_: Thank you for pointing this out. As you mention, NDT3 is a successor of NDT2, and is also a standard Transformer, so there is not much room or perhaps need for technical innovation in the architecture. The primary difference is that NDT2 uses MAE (Masked-autoencoder) dropout of its neural inputs, while NDT3 is a next-step model. A list of more specific technical advances over NDT2 is provided in C.6, but we agree that a detailed comparison of model preparation would be useful, and we have added this in C.3.
>
> _Weakness - Pretrained NDT2 2, Baselining_: We did not collect pretrained NDT2 results as it was designed to show feasibility of scaling through unsupervised learning, not decoding, and from-scratch NDT2 under-performed from-scratch NDT3 on our evaluation datasets, providing no reason to believe that the pretrained NDT2 would perform better than pretrained NDT3. We agree though, that it provides important context, and have added a pretrained NDT2 evaluation. It did not outperform NDT3 from scratch.
>
> _Weaknesses - Novelty of Generalization_: We agree, certainly, the topic of pretraining large models is active work in many subfields of neuroscience. However, different neural data modalities serve widely different communities at this point in time, and spiking data is central enough so as to have a significant audience if a foundation model were successful. Further, Zhang et al’s study of generalization appears most like Sec 3.1, showing broad generalization of the pretrained model when fine-tuning to different downstream tasks. However, Sec 3.2 and 3.3 differ from this general foundation model narrative in that we study generalization of fine-tuned representations, in that we evaluate generalization downstream.

---

> > ### Comment · Reviewer_USSQ · 2024-11-27
> >
> > I thank the authors for discussing the differences with ND2, for elaborating on their goals, and for adding the results on NLB, showing that NDT3 underperforms there and explaining that this is due to its focus on behavior. I also thank them for their discussion of why pretrained NDT2 is not primarily for behavior decoding. I think these revisions strengthen the manuscript. However, as the authors agree, the technical novelty and empirical benefit in this work are limited. They clarify that, instead, their main goal is to demonstrate a generalist model that performs many tasks well. I appreciate the value of this, but this finding with large-scale pretraining is not new, even in this subdomain (though as the authors correctly mention, this is shown for another neural modality). Overall, it remains open whether the scaling trends seen here are specific to NDT3 or not, how this pretrained model would compare to others (e.g. Azabou et al’s POYO, or MtM), and what new insight is gained here to guide the field. Baseline comparisons to other foundation models would have been very helpful, but the authors mention this is not possible because MtM is contemporary/recent work (though weights are available) or POYO’s weights are not available. I think comparison to at least MtM would be possible in the future, given the released weights, and doing so would be very insightful for the field moving forward.

---

> > > ### Author Response · Authors · 2024-11-27
> > >
> > > Thanks for your thoughts. We'd like to discuss your concerns further.
> > >
> > > This study's goal is to analyze how a near-standard Transformer pretrains over broad spiking activity. Evaluating NDT3's *scaling* is only one of three contributions of this work, and the other analyses of NDT3's generalization apply independently of the achieved performance. The **concrete guidance** that these experiments provide for future work is that specialist architectures would be justified if they either outperform NDT3 empirically by saturating beyond 90 minutes across many datasets or outperform NDT3 "theoretically" by generalizing more robustly under input-output manipulations. If you find these analyses valuable but underemphasized, we are open to revising the narrative to highlight them.
> > >
> > > Your main critique that the conclusions may not apply in broader settings is valid. However, this criticism can always be made, if not on more baselines as in our work, then at larger data scales or on more tasks in others. We believe that using limited scope to justify your current score/rejection sets inactionably high standards for our work, as detailed next:
> > >
> > > - **Exhaustive comparison to pretrained models**: Fundamentally, single model comparison is unlikely to provide the insight we both are interested in — whether NDT3’s scaling trends generalize — because we would be comparing a data point to a data trend. Alternatively, reproducing other methods in our pretraining experiments while controlling data/model size would be valuable but is both intractable resource-wise and moreover would yield a different study (breadth across models, over depth in NDT3).
> > > - **Inefficiency of internal comparisons**: One reason we chose to focus on analyses as opposed to baselines is because performance differences between existing models could come from not just the approach, but choice of pretraining datasets, hyperparameter selection, and researcher effort. Disentangling which factors are at play, even when comparing two internally controlled models, is difficult, and is even more so when comparing to externally pretrained models. Therefore, we believe additional pretrained model comparison would not help. Note that we have reported on the appropriate external benchmark (FALCON) to aid future comparison.
> > > - **Moving goalpost**: Prior point aside, we agree that baselines provide useful context. Thus, when you requested comparison to existing models, citing NDT2, POYO, and MtM, we provided results for pretrained NDT2. We chose NDT2 as the tractable and relevant representative, as POYO is not available, and MtM is a model trained on mouse decision-making (that was released 2 months ago and only differs methodologically from NDT in masking). You stated comparison with NDT2 strengthens our work, but now, without changing your score, additional justification for insights gained, or acknowledgement that the other models are unreasonable to compare to, you are requesting further comparisons at the end of revisions.
> > >
> > > Given these, could you clarify if you still believe our work is limited by insufficient comparison?
> > >
> > > We would also like to defend our work’s empirical strength and novelty.
> > >
> > > - **NDT3 does have empirical gain**: Our work states that NDT3 scaling only provides positive transfer when downstream data is under 90 minutes. We framed this conservatively to counteract widespread enthusiasm for large scale pretraining. However, this limit is identified for rigor, and NDT3’s limits exist well beyond the <10 minute downstream setting used in NDT2/POYO. Framed in that setting, the main result in Fig 3D is that scaling *does* improve performance.
> > > - **Generalist novelty**: We appreciate your broad context on neural data models. However, the audience and application for spiking activity/iBCI significantly deviates from those of sEEG and other modalities of neural data. For example, Zhang et al's BRANT line of work targets generalization to EKG (heart biometrics) rather than other neural data modalities. In that sense, their work does not diminish NDT3's novelty any more than, say, [Apple's biometric foundation models](https://arxiv.org/abs/2312.05409).
> > >
> > > Overall, your concern that NDT3's trends may not generalize is valid and important, but advocacy for more pretrained model comparison would not resolve this issue, and we believe that truly resolving this issue will require several future studies. To that end, we believe our work provides a substantial initial analysis that future studies can build upon. If you agree, we hope you might consider revisiting your assessment.

---

### Official Review · Reviewer_xNEo · 2024-11-02

**Soundness:** 2
**Presentation:** 1
**Contribution:** 3
**Rating:** 6
**Confidence:** 3

**Summary:**

This paper introduces a large-scale model that has been pretrained on a large corpus of spiking activity collected from 30 monkeys and humans across 10 labs during diverse motor tasks. It presents a multimodal transformer that tokenizes both neural activity and behavioral data into token streams, allowing the causal decoding of behavior tokens from neural tokens. The findings highlight the challenges of scaling across diverse neural datasets, showing that increasing both model size and data volume can prevent degradation during pre-training. Additionally, the paper suggests that scaling is constrained by variability in input-output relationships, which pre-training alone may not fully resolve.

**Strengths:**

1. This paper attempts to address an important question of whether pretraining large-scale models can yield field-wide improvements for neuroscience.
2. While the model architecture itself may not be novel, there is a lack of effective large-scale multimodal models in this field, and this paper effectively fills that gap.
3. The comprehensive evaluation of how pretraining affects the decoding of diverse motor tasks from datasets of humans and monkeys offers valuable insights for the neuroscience community.

**Weaknesses:**

1. The author observed scaling difficulties, suggesting they may come from variability in datasets. However, I think there are other factors worth considering. For example, model performance may saturate when there is not enough information about behavior in the neural data, and the gain provided by pre-training has less room for improvement. Another factor may be that different behavior modalities/tasks have different training dynamics s.t. the model selected for final evaluation may be good at some tasks but not the others. Thus, it may not be that the model cannot scale across heterogeneous datasets, but that the model has trouble balancing the optimization of different modalities/tasks.
2. Although I don't assess a paper based on its writing and figures, I feel that clearer writing and more understandable figures could better deliver the paper's message. These aspects do not impact the overall quality of the work, but would certainly make it easier for readers to grasp the content. For example, in Fig 3C, it's hard to distinguish different pre-training volume by colors. In Fig 3E, what do the different colors mean?

**Questions:**

I will probably increase my scores if the author can address the questions below:
1. This paper argues that pretraining may be constrained by inherent variability within the neural datasets. Did the author explore other possible explanations for this issue? Given that the model is evaluated on decoding tasks, could it be that certain tasks are inherently easier, leading to performance saturation with little room for improvement? For example, if there is only a limited amount of decodable information about behavior in the neural data, then scaling can not allow us to go beyond that information upper bound. **This is just a hypothesis that is hard to test empirically, but could the author suggest a specific analysis or experiment to test this hypothesis about information upper bounds about some tasks in the neural data?**
2. The author mentioned that “all behavior is normalized per dataset so that the maximum absolute value of each variable is one.” I wonder whether this preprocessing step might influence model training. For example, the small values of all behaviors could lead to a low loss value, which might hinder backpropagation and parameter updates, preventing the model from effectively learning to optimize behavior prediction. Could this be a contributing factor to the scaling challenges? Additionally, why doesn’t the author consider alternative preprocessing methods, such as using the mean and standard deviation to balance the behavior tasks? **Could the authors provide justification for their chosen normalization method, or to conduct an ablation study comparing different preprocessing approaches?**
3. In the evaluation section, the author stated, “To manage compute and storage demands and to reflect that real-world datasets are rarely collected or analyzed in isolation, we fine-tune NDT3 jointly over data combined from multiple evaluation sessions.” I wonder how much this fine-tuning process affects the scaling challenges observed. If the pre-trained model is fine-tuned on multiple, heterogeneous test sets, could this complicate the fine-tuning results? What if the author selected specific test sessions and behavior tasks for targeted fine-tuning? Would that alleviate the scaling difficulties? **Could the author do a simple experiment to quantify the impact of joint vs. targeted fine-tuning on scaling performance?**
4. What specific criteria did the author use for model selection? A common challenge in multimodal multi-task training is that data from different modalities contain varying amounts of information, and some decoding tasks are inherently easier than others. As a result, when one behavior decoding task reaches optimal performance, other prediction tasks may suffer from overfitting. Could model selection contribute to these scaling difficulties? It’s possible that the pre-trained model has already learned to decode each behavior very well during training, but the selected model for final evaluation may be good at certain tasks but bad at others due to imbalanced optimization. **Could the authors provide more details on their model selection process and consider conducting an analysis of task-specific performance across different model checkpoints to investigate potential imbalances?**

---

> ### Author Response · Authors · 2024-11-14
> **Response to xNEo**
>
> _Overall_:
>
> _Weakness - Other factors_: These are interesting points. We discuss them in their corresponding questions (Q1 and Q3) below.
> _Weakness - Clarity_: We appreciate this feedback. We have decreased the number of models reported in the main panels in Figure 3 and believe the colors are now easy to distinguish. The colors throughout Figure 3 now use the same palette, so they still refer to pretraining volume. We are happy to receive other feedback on writing and figures.
>
> _Q - Upper bounds_: As you mention, there is no clear way to test the info-theoretic limit on possible decoding performance. Good deep networks are the current empirical strategy for estimating this limit. We want to distinguish two types of info-theoretic limits. First, there is the [Bayes error](https://arxiv.org/pdf/2110.02095) due to variability in a given neural dataset, timepoint by timepoint, which we believe you are referring to. This is a limit similar to failing to classify an image because it is too blurry. We can estimate this upper bound by scaling training data in our target distribution. For example, this “task-inherent difficulty” cannot be the issue limiting our performance because scaling downstream data continues to improve performance on held-out timepoints in all models. We have calibrated the size of the fine-tuning datasets in our own evaluations so that we are not saturating them due to Bayes error. On the benchmark evaluation of FALCON M1, in contrast (Fig 11, rightmost panel), decoding performance in many models are essentially equivalent.
>
> Second, there is a more subtle limit in how quickly a generalist can align to a test setting. Because the generalist must perform many tasks, we might not be able to be more data efficient in fine-tuning because of variability in the space of potential downstream tasks. This is why some tasks need lengthy prompts for language models to perform well, and is related to the task posed to toy Transformers performing in-context learning of linear regression ([Garg 22](https://arxiv.org/abs/2208.01066)). If the observations are noisy, 2 points may not be enough to fit the line. A final reference is a discussion of how Transformers can be shown to be computing this task recognition on this post by [Adam Shai](https://www.lesswrong.com/posts/gTZ2SxesbHckJ3CkF/transformers-represent-belief-state-geometry-in-their), which mentions computational mechanics as a formal framework for this problem. We have discussed this distinction qualitatively in the introduction in L68-78 and at the end of Sec 3.1.
>
> If the reviewer is interested in a specific experiment to understand how different models overcome this second limit, we would recommend using the cleaner toy settings as studied in Garg 22. Neural data adds the additional complexity of unknown, temporally evolving noise.
>
> _Q - Normalization_: We chose to max-norm over z-scoring because our data is not always unimodal. Our human data often has bimodal behavior (a cursor click signal being off or on). Further our data sometimes contain extreme outliers from sensor errors, which we want to ensure doesn’t disrupt model training. Normalizing by max value would effectively discard these points, while z-scoring is less robust to such outliers. We do not believe this choice impacts scaling, as normalizing data to a fixed small range is standard in machine learning, as long as the loss scale is reasonably adjusted. In practice the dynamic range of the normalized data is spanned by most points in pretraining and evaluation, so we also do not believe this choice of normalization impacts task balance. We are working on plots to demonstrate this point. Further, though this doesn’t exactly address your concern, NDT2 uses z-score normalization and our method compares favorably in direct comparison.
>
> (Response continued in thread)

---

> > ### Author Response · Authors · 2024-11-14
> >
> > _Q - Joint vs Target_: This is an important concern, though we believe Fig 12 (11 in old PDF) already compares a very similar setup, since we use a single test session there. The only difference from your request is that we do “sequential” tuning vs targeted fine-tuning on single-sessions. Sequential fine-tuning differs from the targeted fine-tuning that you ask for in that instead of completely discarding the data from other sessions, we first tune on those data before finally tuning on the final dataset, which achieves a higher performance as in ([Gururangan 20](https://arxiv.org/abs/2004.10964)). Fig 12A reconfirms that tuning for a specific target day alone (left-most point at “0-hours” on the x-axis) performs worse than if we jointly tune with data from other days (other points in panel 12A Cross-Session).
> >
> > Fig 12 previously already showed that joint tuning is not systematically different from sequential tuning when the tuned data when models are pretrained and downstream data is heterogeneous (cross-subject and shuffled conditions). We’ve now added the comparison between joint and sequential tuning for cross-session data, which is closest to our standard evaluation setting. Here, sequential models uniformly perform slightly worse than the jointly tuned models. Thus we conclude that joint fine-tuning is a reasonable method and not limiting our scaling trends.
> >
> > _Q - Task imbalance_: We chose model selection according to best validation loss on our pretraining data, where our validation split is randomly sampled from pretraining (not biased for any given evaluation). We believe this is a reasonable default that matches standard practice for large scale model pretraining. We agree with your concern that some tasks may fit and possibly overfit earlier than others. In general, dynamics in different downstream tasks are expected to vary, as shown in [Pythia](https://arxiv.org/pdf/2304.01373) and [Yang 24](https://arxiv.org/pdf/2404.01204). However, we are unclear how this imbalance might be resolved without new methods, and we would appreciate any references the reviewer has on this topic. Could the reviewer clarify if there is specific insight they hope to gain from evaluating imbalance in NDT3 specifically? We have not saved in-progress checkpoints and so would need to redo pretraining to perform this analysis.

---

> > > ### Comment · Reviewer_xNEo · 2024-11-20
> > > **Thank you for the response**
> > >
> > > Thank you for the response. Q2 and Q3 have been addressed. About Q4, I’m interested in understanding how this imbalance could impact the scaling conclusions of NDT3. The paper states that "NDT3 is trained with mean-squared error for predicting behavioral variables, and categorical cross-entropy losses for predicting neural spike count and reward." Although I appreciate the focus on decoding, I am curious why the test performance on spike prediction is not reported, considering the model’s generalist nature. Could the authors conduct a simple scaling experiment comparing, for example, 1 session to 5 or 10 sessions? It would be helpful to save the best checkpoint for each task and evaluate the model based on those. This could shed light on if there is an imbalance between the behavioral and spike prediction tasks and how this might affect the scaling results.

---

> > > > ### Author Response · Authors · 2024-11-21
> > > > **Follow-up**
> > > >
> > > > Thank you for the elaboration!
> > > >
> > > > For Q4, on multitask mixing, we've added a few analyses to address your concerns. It was unclear whether you were more concerned about multiple downstream tasks improving at difference rates in pretraining, or neural data vs behavioral objectives progressing at different rates in pretraining.
> > > >
> > > > For how downstream overfitting might occur through pretraining, we've retrained a 25-hour NDT3 and evaluated on the bimanual task through this pretraining. The benefit from pretraining is established by 20 epochs into training and persists without overfitting. This plot is shown in B.9.
> > > >
> > > > For the second interpretation, we've added pretraining curves in B.8 that compare the progress of the neural objective (spike prediction) vs the behavioral objective in the pretraining of our main experiments. Other than for the smallest model, overfitting does not occur on the neural data task. You may also appreciate B.3, which shows that pretraining with the neural objective alone still yields scaling, but that using both neural and behavioral objectives is more performant. We did not emphasize reporting on the neural objective as it does not scale as clearly as behavior, and we wanted to defer more thorough investigation to future work.
> > > >
> > > > Finally, we'd like to confirm whether you are also satisfied with our answer to Q1. We've explained how the fact that downstream data scaling improves decoding implies that Bayes' error / saturation in neural data signal cannot be limiting the benefit of pretraining scale. Saturation in neural data signal can occur in some simple behavioral tasks once recording passes a few hours, like in M1. The other aspect of saturation, the saturation in a generalist's ability to tune to new datasets, could be first studied in toy settings like in-context learning of linear regression (Garg 22). We believe that attempting to study this generalist saturation for neural data would only be justified after future works in the field arrive at similar conclusions.
> > > >
> > > > We hope that our additional analyses and responses have addressed all your questions.

---

> > > > > ### Comment · Reviewer_xNEo · 2024-11-22
> > > > > **Thank you for the response**
> > > > >
> > > > > Thank you for your response; I've raised my scores. I appreciate the author's effort to address the questions, though I believe some areas could use more thorough exploration and experimentation. Overall, I think it's important to consider additional factors, such as model architectures and training procedures, before drawing conclusions about scaling. For instance, POYO has demonstrated successful scaling, while the results from NDT3 contradict those of POYO. Furthermore, NDT3 does not perform as well as NDT2 on the FALCON benchmark.

---

> ### Author Response · Authors · 2024-11-22
> **Further clarifications**
>
> We appreciate the raising of the score. We agree that there is much to explore, possibly enough for several future works. We did not wish our paper to overclaim what scaling may look like for neural data, only to start the conversation. If there is a place where you find our language is too broad in claiming scalability for neural data overall, please let us know, and we will revise.
>
> We'd also like to clarify your comments on NDT3's results vs POYO and NDT2. NDT3 evaluates scaling to the 2000 hour regime on 8 tasks, and we have highlighted trouble in a subset of tasks when moving from 200 to 2000 hours. POYO and NDT2 have both demonstrated successful scaling under the 200 hour regime, and specifically only for 2D movement, so we do not believe our work contradicts any existing result. Further, NDT3 handily outperforms NDT2 on the FALCON tasks - please see Figure 11. The NDT2 settings used in our evaluation largely match the publicly recommended settings, so we believe our evaluation to be fair.

---

### Official Review · Reviewer_cs3p · 2024-11-04

**Soundness:** 3
**Presentation:** 2
**Contribution:** 2
**Rating:** 6
**Confidence:** 4

**Summary:**

This paper introduces NDT3, a foundation model for motor decoding from spiking activity. NDT3 is trained on large scale and diverse motor tasks and subjects, demonstrating benefits of scaling law in some scenarios. The paper also reports cases where the model fails to generalize to downstream tasks.

**Strengths:**

* The NDT3 model is original, extending NDT2 with architectural changes to accommodate pretraining on large scale and diverse datasets.
* The paper shows extensive experiments to evaluate the model and was open to discuss failure modes of the model.
* The method is well motivated. Building foundation models for neural data has great implications for BCI and neuroscience applications in general.

**Weaknesses:**

* My main concern about the paper is its lack of clarity in writing. The paper seems to focus a little too much on technical details and verbose explanation, making it hard to follow the main points sometimes. For example, lines 371 to 376 briefly mention joint tuning and sequential tuning, which are minor technical details of fine-tuning choices and do not largely contribute to the main point being discussed in the paragraph which is about input order sensitivity in cross-subject transfer. It might be better to show one type of tuning and leave the other in the Appendix to maintain the flow of writing, which would also help make the plots in Figure 4 cleaner with key takeaways only. Another example is Figure 3C to 3E, where it might be better to only show the 1.5hr, 200hr and 2khr models to avoid cluttering in the plots while still convey the main points.
* Cross-subject transfer which is the main use case of neuroscience pretrained models does not seem to be promising with the proposed model. The usefulness of such foundation model to the community therefore might be limited.

**Questions:**

* Figure 1: are non-neural tokens removed entirely or replaced by zeros tokens? If they are removed entirely from the sequence, would neural tokens assume new positional embeddings left by the non-neural tokens?
* Line 156: why wouldn’t segments be padded with zeros rather than being concatenated with another partial segment to form the two-second cut?
* Figure 3D: what the errorbars would look like if plotted? Pretrained models seem to not differ much by their mean $R^2$. In the absence of errorbars, it’s hard to know if it’s worth scaling up the data and model size for additional performance if the differences are not statistically significant.
* Line 181: why cross-entropy loss was used instead of Poisson likelihood loss that is traditionally used for spike counts? Is there a coefficient balancing MSE loss and cross-entropy loss?
* Figure 5D: was the test monkey with S1/FEF/MT recordings seen during pretraining, i.e. the S1/FEF/MT data might have been recorded from a subject whose motor cortex recordings were present in the pretraining set? If not, this would be the case of cross-subject transfer?

---

> ### Author Response · Authors · 2024-11-14
> **Response to cs3p**
>
> _Overall_: We thank the reviewer for their responses. Given several comments on lack of clarity, we especially appreciate your suggestions for specific figures.
>
> _Weakness - Lack of clarity_: We agree the writing is currently dense and perhaps overly technical. We have updated both figures according to your suggestions. We were concerned about technical completeness in Figure 4 (indeed the specific issue on joint vs sequential tuning is asked about by reviewer xNEo), but we appreciate that this was distracting in the main text. We believe these higher level comments are valuable feedback and would welcome pointers to other places where the flow is distracted.
>
> _Weakness - Limited utility_: We agree that cross-subject use is a primary goal for NDT3. Our evaluations were designed entirely around cross-subject evaluation. Our primary evaluation shows that downstream benefit is variable but overall positive. We believe that this implies the potential for off-the-shelf improvements in at least some portion of the BCI community. On the other hand, we agree NDT3 may not be ready to be a general purpose foundation model for the field, but do believe our experiments highlight important challenges that might be informative for the Neuro-ML comunity.
>
> _Q - Non-neural tokens_: In inference, behavior input tokens are zeroed and phase and return tokens are removed entirely. This removal doesn’t impact positional embeddings of the remaining tokens, as position embeddings are identifying only within each timestep (rotary embeddings are used across time), and to that end, specific ranges of position IDs are reserved for data of different modalities. For example, positions 10-20 are reserved for neural data, position 21 is reserved for return, and positions 21-36 are reserved for behavior. This factorized embedding strategy is described in Sec C.6. We have updated Fig 2 to distinguish the zero-ing vs dropping of tokens.
>
> _Q - Zero-padding_: We concatenate segments to increase the throughput of real data seen in the model. This strategy is standard in language model training and we mention in C.2 that Geiping and Goldstein 2022 has shown this preprocessing choice has little performance impact in language. A full comparison in the neural data domain would be helpful but would require training with the more expensive padded preprocessing.
>
> _Q - Statistics_: Thank you for pointing this out. We’ve now measured significance using paired t-tests with FDR correction, finding that the largest model has statistical significance over all other models except the 350M 200 hr model. Differences between other pretrained models are largely nonsignificant. We’ve highlighted some tests now in the main text and included a full significance matrix in the appendix (Fig 13). Since we are aggregating tasks of widely different performance (e.g. all data points in Fig 1B), we’ve omitted the resulting wide CIs to avoid confusion. We believe this omission is standard in multitask evaluations. For example, on page 46 of the PaliGemma vision-language [model report](https://arxiv.org/pdf/2407.07726), extensive experiments yield average performance benefits across tasks on the order of 2-3%. Nonetheless, CIs derived from these data also overlap much like our own. We would welcome feedback from the reviewer on how to present this issue if they have insights.
>
> _Q -  Cross-entropy over Poisson_: In early experiments, we found this choice did not greatly impact our evaluations and chose the cross-entropy since it is a more broadly common and expressive model, even if Poisson losses are common for neural data models. We mention our existing justification of this choice in C.6: an ablation in Fig 8C shows that ablating the neural loss entirely has minimal impact on evaluation performance. Due to this result, we did not specifically ablate the choice to use the cross-entropy loss. However, we have run a new scaling analysis to support that the cross-entropy loss is nontrivial, labeled Fig 9 in the revision. We tested pretraining with only the unsupervised cross-entropy loss up to 200 hours of data, and fine-tune on downstream tasks, still in a supervised fashion. The fact that there is scaling of downstream performance implies the neural objective is useful. The MSE behavioral objective may obscure differences in the neural objective, though. We’d recommend future work studying specifically neural representation learning, as opposed to decoding, explore this choice in more detail. Further, yes, there is a coefficient to balance MSE and cross-entropy loss to roughly equal orders of magnitude. We experimented with this briefly and found no consistent impact within an order of magnitude of changes on these coefficients.
>
> _Q - FEF/MT_: Yes, throughout our work, monkey evaluations were conducted on subjects held out from pretraining entirely, with the intention of evaluating cross-subject capability. This includes the S1/FEF/MT experiments.

---

> > ### Comment · Reviewer_cs3p · 2024-11-28
> >
> > I thank the authors for the thorough response. With regard to the balance of the MSE and cross-entropy losses, I'm wondering why the cross-entropy loss is necessary during *fine-tuning*. Since the objective of NDT3 is on behavior decoding (measured by the $R^2$ metric), including the cross-entropy loss on predicting neural activity can potentially cause conflict with the main decoding objective. Could the authors comment on how including the cross-entropy loss would benefit model fine-tuning as opposed to using the MSE loss alone? Figure 9 seems to highlight the usefulness of cross-entropy loss in pre-training only.
> >
> > I'm also wondering what the individual loss curves look like during pre-training/fine-tuning, as the model might prioritize optimizing the MSE loss much more than the cross-entropy loss (or vice versa). Thus although the coefficient can keep these losses roughly balanced at the early epochs, these two losses might soon deviate from each other in later epochs. Seeing which loss curve plateaus while the other keeps decreasing might give some hints on its contribution to the training process.
> >
> > Regarding Figure 2, it seems to me that NDT3 is not performing *future* timestep prediction as the term "causal" implies, for it only predicts the next token in the token stream, which are mostly contained within a single timestep except for the Bhvr 2 $\rightarrow$ Neur 1 at the edge. I understand the term "causal transformer" or "causal decoding" is borrowed from the Natural Language Processing literature; however, using these terms here could be misleading, as these tokens do not represent the flow of time. A different term, e.g. autoregressive, might be a better choice, or some clarification should be noted in the text to avoid potential misunderstanding.

---

> > > ### Author Response · Authors · 2024-11-28
> > >
> > > We're glad you appreciate the discussion so far.
> > >
> > > **Cross-entropy in tuning**: In our view, neural cross-entropy serves as a regularizing, auxiliary objective in fine-tuning, same as the role it plays in pretraining. We were motivated by the principle of [Finetune Like You Pretrain](https://arxiv.org/abs/2212.00638) (though the work has a different context), and had anecdotally found using the neural objective was useful in early experiments. We just ran brief experiments re-verifying this on the bimanual / self-paced reaching tasks, where we compared using and not using the neural objective in the downstream task. For pretrained models, the neural objective accelerates tuning and yields mild to large improvements. For from-scratch models, the neural objective also mildly improves performance, though it slows initial learning. We'll add these plots to the appendix.
> > >
> > > **Individual Loss Curves**: We agree that tuning objective weights may be of interest in the future. For this work we roughly balanced the losses so the magnitude of their change over pretraining was comparable (0.01-0.04). In Fig. 14, you can see both individual loss curves roughly continue learning and are not converged by end of pretraining (reviewer xNEo also asked about this). In not-shown pretraining experiments run after the main submission, we amplified the behavior loss by an order of magnitude, but this did not cause robust changes.
> > >
> > > **Causal vs Autoregressive**: Yes, this is a good point. We used causal as a simple functional term to indicate NDT3 does not depend not ever use future data to make predictions, i.e. the model is suitable for realtime BCI control. The model is indeed autoregressive at the token level, and this is the better technical term. We will clarify this in our main approach description.
> > >
> > > If you have no further questions, we would appreciate if you briefly viewed the new figures to judge whether clarity is still a pressing weakness, and to revisit your initial score if you are satisfied. We appreciate the feedback given, regardless.

---

> > > > ### Comment · Reviewer_cs3p · 2024-12-03
> > > >
> > > > I thank the authors for their explanation. I appreciate the large amount of data and analysis the authors put into the study. I am dampened however by the limited cross-subject generalization of the model, which is the most important use case of such pretrained model in practice. It doesn't seem to me that the limited gain could justify the computation overhead in using such large model. Regardless, I think the extensive analyses in the paper offer some valuable insights for the community.

---

### Author Response · Authors · 2024-11-14
**Overall Initial Response to Reviewers**

We appreciate the reviewers’ careful feedback and look forward to a productive discussion period. Globally, reviewers appreciated NDT3’s unprecedented scale and comprehensive experiments. Individually, reviewers commented on NDT3's good motivation and insightful analysis. Critiques focused on three main points:

- _Clarity of writing_: There were concerns with the clarity of writing, with specific feedback concentrated in experiments (Sec 3). We have integrated existing feedback, updating text and Figures 3 and 4. Acknowledging that improving clarity should go beyond explicit feedback, we have updated the text throughout, and particularly in Section 3. Reviewer kGOW has explicitly acknowledged the improvement. We hope that with the reviewers’ feedback we have greatly reduced clarity concerns, and we would be glad to receive feedback on other confusions.

- _Modest benefit of best model_: cs3p, xNEo, and USSQ have noted that our work’s contribution is limited because the gain from pretraining over task-specific experts is small. We agree that the gain, though statistically significant, is limited. However, we would like to point out that NDT3's empirical novelty is mainly in its general competence, rather than advancing state of the art on a specific task. Such a generalist model does not exist in this subfield, with prior models evaluated on 3 datasets at most. Given this first-in-class nature, we expect NDT3, with its limitations, provides the first substantial _scaling_ baseline to justify and guide future work on large scale pretraining on spiking activity. This framing of our study as a guide for future work is supported by the fact that our performance benchmarking is only the first third of our experiments. The other sections provide experiments to support our recommendations for future study of limitations and recommendations for adoption as is.

- _Insufficient baselines_: USSQ and kGOW noted that other works in the field are not compared to in this work. We agree our baselining is sparser than usual, given our focus on analysis. Unfortunately, the proposed baselines either have not released code (EIT) or have not released pretrained weights ([POYO, Azabou et al. 23](https://github.com/neuro-galaxy/poyo/issues/2)), which prevents comparison. As a general note, weak ML reproducibility norms in this subfield have made internal comparisons unreliable, as evidenced by the high variability in method performance reports across the literature. For example, [MtM](https://arxiv.org/abs/2407.14668) reports that NDT2 underperforms NDT1, and POYO reports that [EIT](https://arxiv.org/pdf/2206.06131) underperforms NDT1, conflicting with the respective source works. Controlled benchmarks like the [NLB](https://neurallatents.github.io/) counteract this only when used appropriately. For example, reviewer USSQ suggested comparison with POYO through the NLB, but this is not possible because POYO’s reported results are not visible on the NLB leaderboard, and the NLB is a benchmark on acausal neural representation learning, not (causal) decoding. In our main experiments, we have evaluated NDT3 on the new, directly appropriate FALCON benchmark to assist future comparisons to NDT3. Per reviewer request, we now also report NDT3's performance on the NLB in B.10, though this does not help compare with POYO.

Changes have been posted to the main PDF. Each reviewer has raised important clarification questions, which we address in individual responses. **As of 11/26, we have added the following major updates**:
- We now include pretrained NDT2 as a baseline.
- Fig. 3 (primary evaluations) now include significance testing.
- Fig. 4C-F and corresponding text now shows a reframing of the angular generalization experiments, for clarity.
- Section C.3 now discusses the methodological innovation of NDT3 over NDT2.
- Fig. 9 now provides scaling experiments that pretrain with the neural objective alone.

We are glad 2 reviewers have engaged thus far and improved their rating. We would be glad to clarify further confusions in the discussion period.

---

### Author Response · Authors · 2024-12-03
**End of rebuttal summary**

We thank the reviewers again for their discussion. We summarize key points below:

The reviewer's started with a borderline score, appreciating the work's scale and motivation but listing concerns on writing clarity, insufficient baselines, and insufficient model benefit. For these weaknesses:
- **Clarity**: We have substantially revised the text, with major updates to the experiments (Fig 3/4, Section 3). This was appreciated by kGOW but otherwise not discussed by reviewers.
- **Baselines**: We have added the one viable baseline (pretrained NDT2). Other motor models either have no codebase or weights.
- **Benefit**: Our rebuttal to limited gain is twofold. First, NDT3 provides substantial relative benefit, far outperforming NDT2 in all evaluations. We believe the perception of limited benefit, namely non-benefit on some tasks and beyond 90 minutes of downstream data, comes from the exhaustive rigor of our evaluation relative to prior work. These are at best a critique of the field, as earlier works did not surface these limits because they only evaluated 2D decoding at low downstream timescales. Second, this study's main merit is not that NDT3 provides a ready-to-use foundation model for neuroscience, but that NDT3 provides a baseline account of scaling with a default Transformer, for neuro-ML on spiking activity. While the scoping to NDT architectures may be dissatisfying, this let us focus resources on thorough evaluation. This increases confidence that our account provides the strong baseline needed to justify more specialist architectures and avoid the bitter lesson, given that the similar Transformers work in many domains. Critically, we go beyond acknowledging NDT3's limitations (found early in results) by providing analyses that justify two concrete directions to better understand and potentially overcome said limits.

We are grateful to the reviewers as we believe their feedback has significantly strengthened our work, such that many original critiques have been addressed. Reviewers, have, however, collectively only moved from (5 5 5 6) to (5 6 6 6). We next summarize individual reviews to highlight potential discrepancies:

- **cs3p's** main concern was clarity and secondary concern was weak cost-benefit. Our clarity revisions were not commented on, so could the reviewer please judge whether clarity is still a main concern? For cost, NDT3 is more expensive than prior models but runs in realtime in <8Gb, and so runs on the same consumer GPUs needed for prior models -- we do not believe cost is an applicable critique. For benefit, we have explained why NDT3's benefit may be practically limited but not relative to other works in this nascent subfield, so NDT3 still provides a substantial technical advance. (cs3p score: 6 -> 6)
- **xNEo's** main concern was technical soundness and secondary concern was clarity. We have provided point-by-point feedback or experiments to increase confidence that NDT3's scaling properties do not depend on small methodological choices. We agree that NDT3 does not provide a final account of scaling on spiking activity -- far from it, as NDT3 is only an initial study. Finally, the closing concern on NDT3 underperformance is incorrect, and this remains unaddressed. We wonder whether this correction, combined with clarity revisions, would change your final scoring. (xNEo score: 5 -> 6)
- **USSQ's** main concern was limited benefit and baselining with secondary concern on novelty. We have discussed why limited benefit, while a fair critique, should not overshadow that NDT3 substantially advances this subfield in raw performance and generality of benefit. For baselines, we have added pretrained NDT2 in rebuttal and noted RNN comparisons are available on FALCON (NDT3 well outperforms both). Reviewer USSQ appeared to request comparison with MtM at the end of rebuttals, but we believe this is out of scope given that MtM studies a different domain (cross region, mouse-brain, neural activity prediction) and is less than 2 months old. Relatedly, we hope this explicit contrast within spiking models makes clear that the spiking activity domain is different from sEEG (BRANT2), so that critiques of novelty are less applicable. Overall, we agree with your closing comment that NDT3's scaling may be specific to its methods and not fundamental to spiking activity, but we believe that such an broad claim is beyond the scope of one work. (USSQ score: 5 -> 5)
- **kGOW's** primary concern was limited baselining and secondary concern was clarity. In response to our comments that EIT/POYO are unavailable to compare to, our note of codebase readiness, and improved clarity, kGOW amended their score. (kGOW score: 5 -> 6).

---

### Meta-Review · Area_Chair_x6BS · 2024-12-16

**Metareview:**

This paper presents Neural Data Transformer 3 (NDT3), a pretrained foundation model for decoding motor information from spike-based recordings. The model is a causal transformer model using simple tokenization approaches, pretrained with 2000 hours of neural population spiking activity from over dozens of monkeys and several labs. The authors claim that fine-tuning the pretrained model enables improved decoding and generalization, with better efficiency and accuracy in response to distribution shifts. Finally, the authors note that the scaling behavior of the model is more limited, and they propose that this may be a result of the fundamental heterogeneity in neural data.

The strengths of this paper are: (1) It is well-motivated, and addresses an important question of whether pretraining large-scale models can yield general improvements for motor decoding. (2) The evaluations are extensive. (3) The authors train on a very large dataset, the largest used in motor decoding to date.

The weaknesses are: (1) The clarity of the writing was not great. (2) Limited evidence for improved transfer as a result of increased data scale (which suggests that the model is not fully taking advantage of the scale of the dataset). And, (3) Limited baselines as comparisons. The reviewers raised these weaknesses to the authors.

The authors attempted to address the reviewers' concerns, and were partially successful. The reviewers recognized the improvement in baseline comparisons and clarity. But, the concerns about the actual benefits of the model were not fully assuaged. Given these considerations, and the final scores (average 5.75), a decision to reject was reached.

**Additional Comments On Reviewer Discussion:**

The authors and reviewers engaged in discussion, and the scores did move up. Generally, the discussion was constructive. However, the authors did not fully assuage the reviewers' concerns about the usefulness of the model, given the lack of clear impriving generalization with increased data scale.

---

### Decision · Program_Chairs · 2025-01-22

Reject